# Imitating the Truth: Attention-aware Truth-Guided Enhancement for Hallucination Mitigation in Large Vision-Language Models

**Hairui Ren[1], Zixuan Wang[1], Yibo Yang[2], He Zhao[3 4], Fan Tang[5], Dandan Guo[1 2*],**
**Yi Chang[1 6 7*]**

School of Artificial Intelligence, Jilin University[1]; KAUST[2];
CSIRO's Data61[3]; Department of DSAI, Monash University[4];
Institute of Computing Technology, Chinese Academy of Sciences[5]
International Center of Future Science, Jilin University[6]
Engineering Research Center of Knowledge-Driven Human-Machine Intelligence, MOE, China[7]
`{renhr22,zixuan24}@mails.jlu.edu.cn,`
`yibo.yang93@gmail.com, he.zhao@data61.csiro.au,`
`tangfan@ict.ac.cn, {guodandan,yichang}@jlu.edu.cn`

## Abstract

Large Vision-Language Models (LVLMs) achieve impressive multimodal reasoning but remain prone to hallucinations, generating content inconsistent with visual evidence. Existing mitigation methods often rely on auxiliary modules or coarse decoding-time adjustments, overlooking the fine-grained dynamics that distinguish truthful (real) tokens from hallucinatory ones. In this paper, we introduce **AGE (Attention-aware Truth-Guided Enhancement)**, a training-free framework that performs fine-grained, layer-wise interventions guided by attention patterns of real tokens. Our analysis reveals that real and hallucinated tokens follow distinct stage-specific attention behaviors, and hallucinations emerge when models fail to reproduce these behaviors. AGE addresses this by introducing two lightweight interventions: (i) Imitating the image attention, derived from discrepancies between real and hallucinated tokens, and (ii) Imitating the text attention when semantic grounding is required. Extensive experiments on widely used benchmarks, including COCO Image Captioning, POPE, and MME, demonstrate that AGE consistently mitigates hallucinations across diverse LVLMs such as LLaVA, MiniGPT-4, and mPLUG-Owl2, without additional training or loss of fluency. Our results highlight that imitating truth-grounded attention dynamics is a simple yet powerful principle to improve the reliability of LVLMs.

## 1 Introduction

Large Vision-Language Models (LVLMs) (Chen et al., 2023; Li et al., 2023a; Liu et al., 2023b; Zhu et al., 2023; Ye et al., 2023) have demonstrated remarkable capabilities across a wide range of multimodal tasks, including image caption (Hu et al., 2023b), visual question answer (Liu et al., 2024c), and instruction following grounded in visual content (Hong et al., 2024; Liu et al., 2024a). Despite their impressive capabilities, LVLMs often suffer from a critical flaw: hallucination, the generation of content that is not supported or contradicted by the visual input. This issue poses a substantial threat to their reliability and limits deployment in high-stakes scenarios such as autonomous systems (Chen et al., 2024c; Mai et al., 2023) and healthcare diagnostics (Hu et al., 2023a; Wang et al., 2024b). Understanding the causes of hallucinations in LVLMs and devising effective strategies for their mitigation are essential steps toward enhancing the reliability of these models.

Given the complexity of LVLM architectures and their reliance on multimodal fusion reasoning, eliminating hallucinations remains highly challenging. Recent studies have explored two major directions: introducing external auxiliary modules (Zhou et al., 2023; Yin et al., 2024) and intervening during decoding (Huang et al., 2024; Leng et al., 2024; Yang et al., 2025b). Among these,

---

*Corresponding authors.

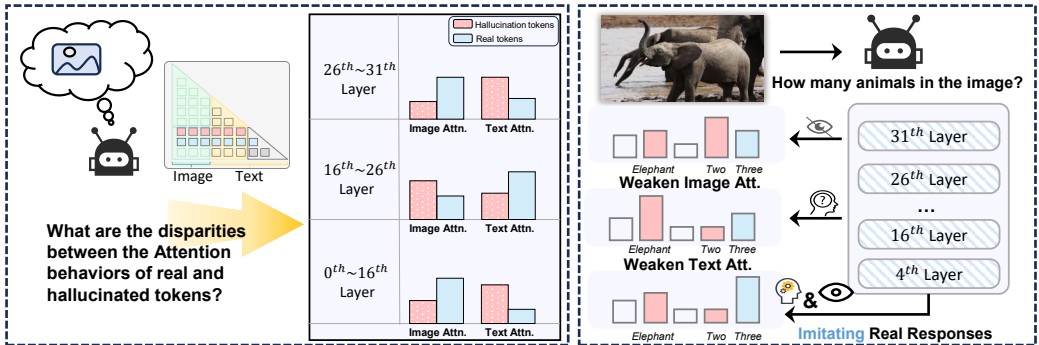

Figure 1: An illustration of our core motivation. Left: The attention behaviors for real and hallucinated tokens exhibit clear, stage-dependent disparities. Right: A conceptual example demonstrating the consequences of these disparities. Weakening the dominant modality at a given stage (*e.g.*, text attention for counting, image attention for final grounding) leads to specific hallucinated outputs.

decoding-time approaches such as OPERA (Huang et al., 2024) and VCD (Leng et al., 2024) have drawn particular attention due to their model-agnostic nature and ease of deployment. These methods typically mitigate hallucinations by reweighting visual attention or counteracting biased text priors during generation. However, most of them (Huang et al., 2024; Chen et al., 2024a; Park et al., 2025; Leng et al., 2024; Zou et al., 2025) operate in a coarse-grained manner, applying uniform enhancements across layers or modalities. Such oversimplification fails to capture the nuanced dynamics of multimodal reasoning, resulting in reduced robustness and less effective hallucination mitigation.

To move beyond coarse-grained interventions, we adopt a finer-grained analysis, where we examine hallucinated responses by decomposing them into **"truthful tokens"** (real tokens image-grounded) and **"hallucinatory tokens"** (unsupported) to analyze their attention behaviors layer by layer. As visualized in Fig. 1 (more details in Fig. 2), this investigation reveals systematic, stage-dependent discrepancies across both modalities: the attention patterns for real tokens, in both visual and textual domains, differ significantly from those of hallucinatory tokens, and this pattern of divergence is both model- and stage-specific. For instance, when the model fails to sufficiently attend to text in the middle stage as a real response would, it may default to a salient visual object (e.g., "Elephant") instead of the correct count. Conversely, insufficient visual attention in the late stage can lead to over-reliance on incomplete textual priors (e.g., generating "Two" instead of "Three"). These findings suggest that hallucinations are caused by a **failure to reproduce the token-level, stage-sensitive attention dynamics characteristic of real responses**. This insight motivates our core hypothesis: hallucinations can be mitigated by guiding the model to imitate the internal attention behaviors of real tokens through adaptive, stage-specific interventions.

Motivated by this insight, we introduce **AGE (Attention-aware Truth-Guided Enhancement)**, a training-free, decoding-time framework that mitigates hallucinations by imitating the stage-specific attention dynamics of real tokens. Rather than applying uniform adjustments, AGE implements targeted interventions aligned with the distinct attention characteristics of each model, focusing on the stages where discrepancies are most pronounced. Concretely, it employs two lightweight interventions that require: (i) **Imitating the Image Attention**, derived from the attention disparity between real and hallucinatory tokens, to restore visual grounding in late reasoning layers; and (ii) **Imitating the Text Attention** where analysis reveals a stronger reliance on text, as observed in the middle stage of LLaVA. By imitating attention behaviors in real tokens, AGE enables more accurate, fluent, and trustworthy multimodal generation. We summarize our primary contributions as follows:

1. We conduct a novel token-level, layer-wise analysis of attention within hallucinated responses, identifying the cause of hallucinations as a failure to reproduce stage-specific attention behaviors of real tokens.

2. We design and propose **AGE**, a framework whose lightweight interventions effectively translate our analytical insights into a practical, decoding-time solution.

3. We provide extensive experimental validation showing that AGE significantly and consistently reduces hallucinations across multiple LVLMs and benchmarks, without sacrificing fluency or completeness.

## 2 RELATED WORK

**Large Vision-Language Models.** The remarkable success of large language models (LLMs) (Touvron et al., 2023a;b) has spurred increasing interest in extending their capabilities to the multimodal domain. With the open release of influential LLM backbones such as LLaMA (Touvron et al., 2023a) and Vicuna (Chiang et al., 2023), large vision–language models (LVLMs) (Bai et al., 2023; Chen et al., 2024b; Liu et al., 2024a; 2023b; Zhu et al., 2023) have rapidly emerged as powerful systems capable of comprehensively processing and generating content across multiple modalities, including text, images, and even audio. Building on these foundations, models such as LLaVA (Liu et al., 2023b), mPLUG-Owl2 (Ye et al., 2023), and MiniGPT-4 (Zhu et al., 2023) further advance interactivity by supporting joint image–text prompts, enabling richer and more context-aware responses. Most LVLMs employ a two-stage training paradigm: an initial vision–language feature alignment phase followed by instruction tuning, which equips them to interpret and follow multimodal queries effectively. However, despite these advancements, hallucination, where generated content diverges from or contradicts the visual evidence, remains a persistent and widespread limitation across current LVLMs.

**Mitigating Hallucinations in LVLMs.** Hallucination, content that is irrelevant, inaccurate, or inconsistent with the visual input (Bai et al., 2024), has been linked to limitations in visual encoding (Tong et al., 2024; Liu et al., 2024b; Shi et al., 2024; Liu et al., 2024d), overreliance on parametric knowledge (Zhou et al., 2023; Leng et al., 2024), and noisy supervision (Liu et al., 2023a; Yu et al., 2024). Mitigation strategies include training-based improvements via cleaner or reweighted datasets (Yue et al., 2024; Jiang et al., 2024) and decoding-time adjustments (Yang et al., 2025a; Zhang et al., 2025; An et al., 2024; Wang et al., 2024a) such as VCD (Leng et al., 2024), which contrasts output distributions derived from original and distorted visual in puts, or attention calibration OPERA (Huang et al., 2024), which penalizes over-trust and refines token selection based on previous outputs. However, most of these approaches require costly retraining, depend on post-generation filtering, or apply coarse global attention changes. In contrast, our method directly calibrates attention behaviors during inference by imitating real response patterns, offering a fine-grained intervention across reasoning stages without modifying model training or architecture.

## 3 UNCOVERING HALLUCINATION-REAL RESPONSE DISCREPANCY

### 3.1 PRELIMINARY

Large Vision-Language Models (LVLMs) process an image $V$ and a textual instruction to generate a response $Y = \{y_1, \ldots, y_K\}$. The image is encoded into visual tokens $\{v_1, \ldots, v_n\}$, and the instruction into text tokens $\{t_1, \ldots, t_m\}$. These models generate text autoregressively using an $L$-layer Transformer decoder. At each decoding step $k$ and layer $l$, the model computes an attention weight vector $\mathbf{a}^{(l,k)} \in \mathbb{R}^{n+m+k}$. This vector is composed of a visual component $\mathbf{a}^{(l,k)}_{\text{vision}} \in \mathbb{R}^n$ over the visual tokens and a textual component $\mathbf{a}^{(l,k)}_{\text{text}} \in \mathbb{R}^{m+k}$ over the textual (instruction and previously generated) tokens. These attention weights are then used to update the hidden state $\mathbf{h}^{(l)}_k \in \mathbb{R}^d$ from the corresponding value matrices ($\mathbf{V}^{(l)}_{\text{vision}} \in \mathbb{R}^{n \times d}$ and $\mathbf{V}^{(l,k)}_{\text{text}} \in \mathbb{R}^{(m+k) \times d}$) via a residual connection:

$$\mathbf{h}^{(l+1)}_k = \mathbf{h}^{(l)}_k + \text{AttentionSubLayer}\left(\mathbf{a}^{(l,k)}_{\text{vision}}, \mathbf{V}^{(l)}_{\text{vision}}, \mathbf{a}^{(l,k)}_{\text{text}}, \mathbf{V}^{(l,k)}_{\text{text}}\right). \quad (1)$$

The probability of the next token is predicted from the final layer's hidden state, $\mathbf{h}^{(L)}_k$:

$$p(y_k|y_{<k}) = \text{Softmax}(f(\mathbf{h}^{(L)}_k)), \quad (2)$$

where $f(\cdot)$ is an affine layer. Hallucination may occur when the generated text $Y$ is inconsistent with or contradicts the image $V$.

## 3.2 METHODOLOGY FOR ANALYZING ATTENTION DISCREPANCY

While prior work often attributes hallucination to a general lack of visual grounding or textual prior interference, we hypothesize that the underlying cause is a more nuanced, dynamic behavior in attention. Specifically, we conduct a fine-grained analysis of the internal attention dynamics of LVLMs to uncover stage-dependent patterns that can inform a more targeted intervention. Our analysis focuses on three representative LVLMs: LLaVA-1.5-7B (Liu et al., 2023b), MiniGPT-4 (Zhu et al., 2023), and mPLUG-Owl2 (Ye et al., 2023). For these models, we curated a set of $\{V_i\}_{i=1}^{N}$ with $N = 100$ challenging images from the COCO training set, specifically selected for their known propensity to elicit hallucinated responses $\{Y_i\}_{i=1}^{N}$. Our core idea is to investigate if a model exhibits different attention behaviors when generating real versus hallucinated content within the same response.

For each generated response $Y^{(i)}$ that contained inaccuracies, we distinguish real and hallucinated tokens by comparing the predicted objects with the ground-truth annotations. A token was labeled as "real" only if it corresponded to an object explicitly present in the ground-truth labels, and otherwise marked as hallucinated. The resulting sets are: (1) **Real Tokens** $I_{\text{real}}^{(i)}$: Tokens corresponding to objects verifiably present in the image; and (2) **Hallucinated Tokens** $I_{\text{hall}}^{(i)}$: Tokens corresponding to objects confabulated by the model and not present in the image. To quantify the difference in attention behavior between these two token sets, we first define the **Average Attention Sum**, $\bar{s}$, a per-sample metric for a specific modality and token type. For instance, the average visual attention sum for real and hallucinated tokens in sample $i$ at layer $l$ can be expressed as:

$$\bar{s}_{(\text{real},\text{vision})}^{(l,i)} = \frac{1}{|I_{real}^{(i)}|} \sum_{k \in I_{\text{real}}^{(i)}} \text{sum}\left(\mathbf{a}_{\text{vision}}^{(l,k)}\right), \quad \bar{s}_{(\text{hall},\text{vision})}^{(l,i)} = \frac{1}{|I_{\text{hall}}^{(i)}|} \sum_{k \in I_{\text{hall}}^{(i)}} \text{sum}\left(\mathbf{a}_{\text{vision}}^{(l,k)}\right), \quad (3)$$

where $\text{sum}(\cdot)$ is the sum of all elements in a vector. The corresponding terms for the textual modality are defined analogously. After computing these per-sample scores, we aggregate them across all $N$ images to derive our final layer-wise difference metrics:

$$\text{Diff}_{\text{image}}^{l} = \frac{1}{N} \sum_{i=1}^{N} \left(\bar{s}_{(\text{real},\text{vision})}^{(l,i)} - \bar{s}_{(\text{hall},\text{vision})}^{(l,i)}\right), \quad \text{Diff}_{\text{text}}^{l} = \frac{1}{N} \sum_{i=1}^{N} \left(\bar{s}_{(\text{real},\text{text})}^{(l,i)} - \bar{s}_{(\text{hall},\text{text})}^{(l,i)}\right). \quad (4)$$

Consequently, a positive value in $\text{Diff}_{\text{image}}^{l}$ signifies that real tokens, on average, allocate greater attention to the visual modality than hallucinated tokens at layer $l$. This finding forms the basis of our intervention: guiding the model to align with the attention patterns of truthful tokens.

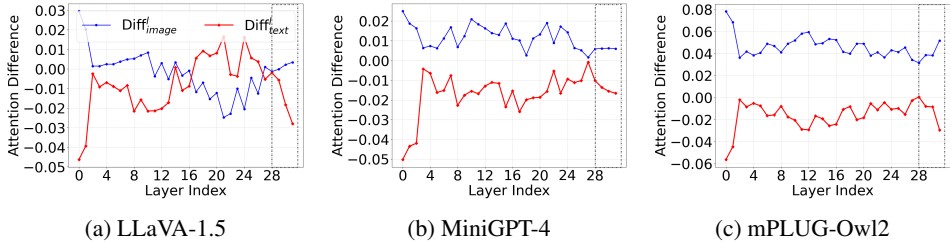

(a) LLaVA-1.5        (b) MiniGPT-4        (c) mPLUG-Owl2

Figure 2: A layer-wise characterization of attention disparities between real and hallucinated responses, which vary across LLaVA-1.5, MiniGPT-4, and mPLUG-Owl2.

## 3.3 ANALYSIS OF ATTENTION BEHAVIORS

As shown in Fig. 2, our analysis of real and hallucinated token sets reveals distinct, model-specific attention behaviors. To facilitate this analysis, we partition each model's architecture into three stages based on the observed dynamics: Early (Layers 0-16), Middle (Layers 16-26), and Late (Layers 26-31). The attention dynamics are notably model-specific. LLaVA-1.5, for instance, reveals a complex non-monotonic relationship where real responses show stronger visual attention in the early and late stages but rely more heavily on textual context in the middle stage. In contrast, MiniGPT-4 and mPLUG-Owl2 present a more consistent dynamic, with visual attention for real responses decisively dominating across nearly all stages. Despite these differences, a critical commonality emerges: Across all tested models, the final reasoning stage (Layers 26-31) exhibits a

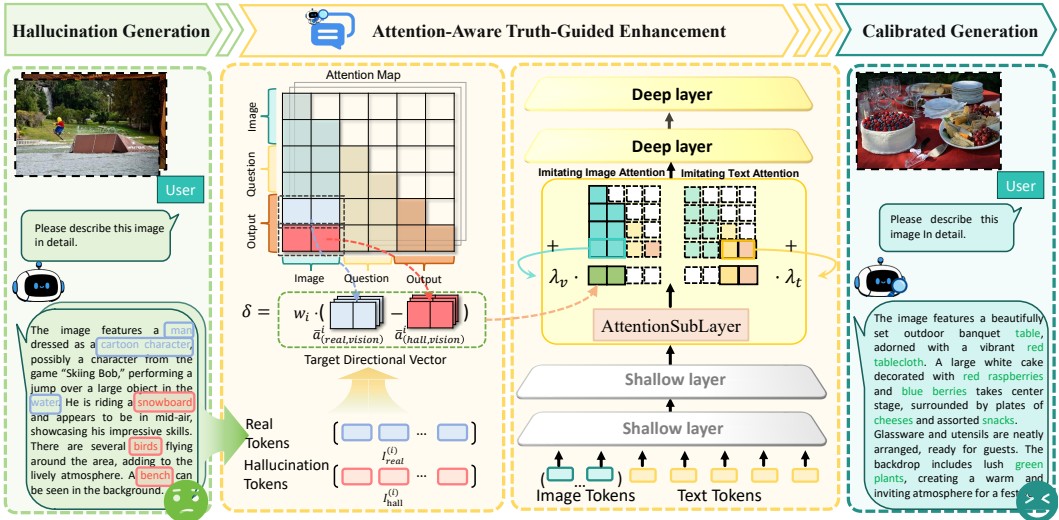

Figure 3: AGE first samples hallucinatory responses and distinguishes real from hallucinated tokens. The visual attention gap between them yields the target vector $\delta$. During inference, AGE applies $\delta$ and/or textual self-multiplicative enhancement, guiding LVLMs to align with real responses and mitigate hallucinations.

stable and pronounced positive gap where real responses attend more to visual tokens than hallucinated ones. This provides a universal and impactful opportunity for intervention. See Appendix A.8 for more results about different size $N$.

This crucial finding demonstrates that hallucination is not caused by a simple, uniform lack of visual attention, but is instead tied to a failure to replicate nuanced, model-specific, and stage-specific attention dynamics. Consequently, naive, one-size-fits-all interventions that globally enhance visual attention are insufficient and may even be counterproductive. Our analysis, therefore, points to a more effective strategy: we hypothesize that hallucinations can be mitigated by guiding a model to imitate the attention behavior of real responses. This requires interventions that are not only targeted to specific reasoning stages but are also adaptive to the unique attention patterns of each individual model. Motivated by this core insight, we introduce a novel framework in the following section, designed to implement this principle through targeted attention interventions faithfully.

## 4 OUR PROPOSED METHOD

This work introduces AGE (**A**ttention-aware Truth-**G**uided **E**nhancement). AGE is a training-free, decoding-time framework that corrects these attention disparities by guiding the model to imitate the attention patterns of real responses.

### 4.1 IMITATING THE ATTENTION BEHAVIOR

**Imitating the Image Attention.** Our analysis in Section 3.3 revealed a critical, universal finding: across all tested LVLMs, the late reasoning stages consistently exhibit a stable pattern where real responses allocate significantly more attention to visual tokens. While a simple self-multiplicative enhancement could increase visual focus, it is direction-agnostic and coarse. This work proposes a directional calibration of visual attention.

As shown in Fig. 3, our strategy is to compute a single, robust directional vector, $\delta \in \mathbb{R}^n$, that captures the essential shift from a hallucinatory to a real attention pattern. The dimension $n$ corresponds to the number of visual tokens produced by the image encoder, as introduced in Section 3.1. To compute this vector, we first randomly select $M$ samples from the COCO training set that are known to elicit hallucinatory responses. The vector $\delta$ is designed to capture a general corrective tendency in the model's attention space, rather than overfitting to the specifics of the $M$ samples. For each sample $i \in [1, M]$, we follow the methodology from Section 3.2 and partition its generated tokens into two distinct sets: real tokens ($I_{\text{real}}^{(i)}$) and hallucinatory tokens ($I_{\text{hall}}^{(i)}$). To quantify the atten-

tion patterns for these token sets, we define the **Average Attention Vector as** $\overline{\mathbf{a}} \in \mathbb{R}^n$. This vector represents the averaged attention distribution over the $n$ visual tokens. It is calculated by averaging the attention vectors from a specific layer for a given set of tokens. We choose to compute this from the final decoder layer ($L$), as it contains the most semantically integrated representations just prior to token generation. For a given sample $i$, the average image attention vector for its *hallucinated tokens* and *real tokens* are formally defined as:

$$\overline{\mathbf{a}}^i_{(\text{hall,vision})} = \frac{1}{|I^{(i)}_{\text{hall}}|} \sum_{k \in I^{(i)}_{\text{hall}}} \mathbf{a}^{(L,k)}_{\text{vision}}, \quad \overline{\mathbf{a}}^i_{(\text{real,vision})} = \frac{1}{|I^{(i)}_{\text{real}}|} \sum_{k \in I^{(i)}_{\text{real}}} \mathbf{a}^{(L,k)}_{\text{vision}}. \tag{5}$$

The target directional vector $\boldsymbol{\delta}$ is then calculated as the weighted average of the difference between these real and hallucinatory vectors across the $M$ samples (see Appendix A.3 for the adaptive weighting scheme):

$$\boldsymbol{\delta} = \frac{1}{M} \sum_{i=1}^{M} w_i \cdot (\overline{\mathbf{a}}^i_{(\text{real,vision})} - \overline{\mathbf{a}}^i_{(\text{hall,vision})}). \tag{6}$$

During inference, we inject this universal adjustment vector into the visual attention of layers within the pre-defined late stage:

$$\hat{\mathbf{a}}^l_{\text{vision}} = \mathbf{a}^l_{\text{vision}} + \lambda_v \times \boldsymbol{\delta}. \tag{7}$$

where $\lambda_v$ is a scaling factor. This encourages the model to reproduce the universal image attention behavior characteristic of truthful responses.

**Imitating the Text Attention.** As our analysis also highlighted, attention dynamics can be highly model-specific. For models like MiniGPT-4 and mPLUG-Owl2, where visual attention dominates throughout, no textual intervention is necessary. However, for LLaVA-1.5, the middle stage shows that real responses rely substantially more on textual context. To address this model-specific need, we reinforce text attention in the designated middle stage of LLaVA. Since text attention vectors dynamically change in dimension, pre-computing a fixed directional vector is infeasible. Therefore, we adopt a self-multiplicative enhancement as an effective proxy to amplify the model's focus on its generated context:

$$\hat{\mathbf{a}}^l_{\text{text}} = \mathbf{a}^l_{\text{text}} + \lambda_t \times \mathbf{a}^l_{\text{text}}, \tag{8}$$

where $\lambda_t$ is a scaling factor. This targeted adjustment ensures our framework adapts to the unique reasoning patterns of each LVLM.

While this method does not specify a corrective direction, it effectively amplifies the model's focus on its generated context, thereby imitating the attention behavior uncovered in our analysis. Unlike prior work, our approach modulates multimodal attention in a stage-specific manner to faithfully reproduce the attention behavior of real responses.

## 4.2 CALIBRATED AUTOREGRESSIVE GENERATION

The stage-specific adjustments described previously are integrated into the standard autoregressive generation process. At each decoding step $k$ and for each decoder layer $l$, the model applies the corresponding intervention conditionally based on the layer's designated stage. Specificall, for all LVLMs, we choose $l$ from Late Stage and shift the visual attention using the directional vector $\boldsymbol{\delta}$ as defined in Eq. 7; for LLaVA, we additional enhance the textual attention via self-multiplication in Eq. 8 by choosing $l$ from Middle Stage. Attention in layers outside these defined stages remains unchanged. The model then computes the subsequent hidden state $\mathbf{h}^{(l+1)}_k$ using these potentially calibrated attention scores, $\hat{\mathbf{a}}^{(l,k)}_{\text{vision}}$ and $\hat{\mathbf{a}}^{(l,k)}_{\text{text}}$, to form the context vector. This is conceptually represented by:

$$\mathbf{h}^{(l+1)}_k = \mathbf{h}^{(l)}_k + \text{AttentionSubLayer}(\hat{\mathbf{a}}^{(l,k)}_{\text{vision}}, \mathbf{V}^{(l)}_{\text{vision}}, \hat{\mathbf{a}}^{(l,k)}_{\text{text}}, \mathbf{V}^{(l,k)}_{\text{text}}). \tag{9}$$

By coordinating these interventions in different LVLMs, AGE calibrates attention behavior throughout the reasoning process guided by the direction of real responses. This design not only suppresses hallucinations but also ensures faithful alignment with visual evidence, providing a principled and interpretable pathway toward more trustworthy multimodal generation. The algorithm of AGE is summarized in the Appendix A.2.

Table 1: Hallucination rates (%) are reported using CHAIR$_S$ ($C_S$), CHAIR$_I$ ($C_I$), and BLEU (%) on COCO image captioning tasks, where lower CHAIR and higher BLEU are better. The $max\ new$ $token$ is set to 64. The best results are highlighted in **bold** while the second-best results are marked with underline. [†] represents the results reported from Chen et al. (2024d). [‡] represents the results reported from the corresponding original paper.

| Method | MiniGPT-4-7B | | | LLaVA-1.5-7B | | | mPLUG-Owl2-7B | | | Avg. | | |
| | $C_S \downarrow$ | $C_I \downarrow$ | BLEU↑ | $C_S \downarrow$ | $C_I \downarrow$ | BLEU↑ | $C_S \downarrow$ | $C_I \downarrow$ | BLEU↑ | $C_S \downarrow$ | $C_I \downarrow$ | BLEU↑ |
|---|---|---|---|---|---|---|---|---|---|---|---|---|
| Greedy [†] | 30.87 | 12.33 | 14.33 | 20.80 | 6.77 | 15.93 | 23.20 | 8.33 | 15.37 | 24.95 | 9.14 | 15.21 |
| Beam Search [†] | 29.56 | 11.36 | 14.94 | 18.67 | 6.30 | 16.17 | 21.67 | 7.63 | 15.77 | 23.30 | 8.43 | 15.62 |
| DoLA (Chuang et al., 2023) [†] | 30.87 | 11.70 | 14.93 | 21.00 | 6.70 | 15.93 | 24.60 | 8.73 | 15.40 | 25.49 | 9.04 | 15.42 |
| OPERA (Huang et al., 2024) [†] | 30.00 | 11.67 | 14.87 | 21.13 | 6.73 | 16.27 | 22.13 | 7.57 | 15.53 | 24.42 | 8.65 | 15.56 |
| VCD (Leng et al., 2024) [†] | 30.27 | 12.60 | 14.33 | 23.33 | 7.90 | 14.67 | 27.27 | 9.73 | 14.40 | 26.95 | 10.07 | 14.46 |
| Woodepecker (Yin et al., 2024) [†] | 28.87 | 10.20 | 15.30 | 23.85 | 7.50 | 17.05 | 26.33 | 8.43 | 16.43 | 26.35 | 8.71 | 16.26 |
| LURE (Zhou et al., 2023) [†] | 27.88 | 10.20 | 15.03 | 19.48 | 6.50 | 15.97 | 21.27 | 7.67 | 15.65 | 22.87 | 8.12 | 15.55 |
| VISTA (Li et al., 2025) [‡] | 19.80 | 6.00 | - | 20.40 | 6.90 | - | - | - | - | 20.10 | 6.45 | - |
| Ours | **15.62** | **6.00** | **15.79** | **16.43** | **5.58** | 16.48 | **19.40** | **7.47** | 16.21 | **17.15** | **6.35** | 16.16 |

## 5 EXPERIMENTS

**Baselines.** We evaluate our approach on three representative LVLMs: LLaVA-1.5 (Liu et al., 2023b), mPLUG-Owl2 (Ye et al., 2023), and MiniGPT-4 (Zhu et al., 2023). To examine the impact of model scale, we additionally experiment with the 13B variants of LLaVA-1.5. We use the default greedy decoding strategy and compare AGE against recently SOTA methods: DoLA (Chuang et al., 2023), OPERA (Huang et al., 2024), VCD (Leng et al., 2024), Woodepecker (Yin et al., 2024), LURE (Zhou et al., 2023), ICD (Wang et al., 2024c), and VISTA (Li et al., 2025).

**Evaluation Metrics.** We assess our method using a suite of standard hallucination benchmarks, in line with prior studies (Huang et al., 2024; Leng et al., 2024). (1) First, for image captioning, we employ the Caption Hallucination Assessment with Image Relevance (CHAIR) (Rohrbach et al., 2018). This involves generating descriptions for 500 COCO validation images and comparing them against ground-truth objects to measure hallucination at both the sentence (CHAIR$_S$) and instance (CHAIR$_I$) levels. To ensure responses remain faithful to the visual content, we also report an instance-level BLEU score (Papineni et al., 2002). (2) Second, we evaluate perceptual hallucinations using the POPE benchmark (Li et al., 2023b), which probes a model's ability to identify the presence or absence of visual concepts through 3,000 binary questions. We report Accuracy and F1 scores across its Random, Popular, and Adversarial settings. (3) Finally, we use the MME benchmark (Fu et al., 2023), focusing on four hallucination-related subtasks: Existence, Count, Position, and Color. Following the protocol of Yin et al. (2024), we report the overall accuracy as the evaluation metric. Please see more details in Appendix A.4.

**Hyperparameter setting.** In our experiments, interventions are applied at the $20th$ layer for the Middle stage and at the $30th$ and $31st$ layers for the Late stage. The scaling factors are set to $\lambda_v = 100$ for visual attention and $\lambda_t = 3$ for textual attention. The number of $M$ (samples to generate $\delta$) is set to 10. Please refer to the Appendix A.7 for more details.

### 5.1 MAIN RESULTS

**Results on COCO Image Captioning.** To evaluate the effectiveness of our method on the captioning task, we adopt the CHAIR metric on COCO. Specifically, we conduct experiments on three representative models: LLaVA-1.5-7B (Liu et al., 2023b), MiniGPT-4 (Zhu et al., 2023), and mPLUG-Owl2 (Ye et al., 2023). As shown in Table 1, our method outperforms the latest state-of-the-art (SOTA) method by a margin of $2.85\%$ and $0.10\%$ on CHAIR$_S$ and CHAIR$_I$, respectively, demonstrating that our AGE substantially mitigates hallucinations during multimodal reasoning. Importantly, our approach maintains comparable BLEU scores to the baseline models, even a $0.95\%$ improvement, indicating that the mitigation in hallucination is not achieved at the expense of response quality. It is worth noting that both LURE (Zhou et al., 2023) and Woodpecker (Yin et al., 2024) rely on additional training data or auxiliary models to mitigate hallucinations. In contrast, AGE achieves superior performance using a minimal set of only 10 images. This highlights that the improvements of our approach do not stem from external data augmentation, but rather from faithfully replicating the nuanced, stage-specific attention dynamics observed in real responses by imitating their attention behaviors. Please refer to the Appendix A.6 for more results about different sizes of models.

**Results on POPE.** We further validate our method on the POPE benchmark. Following the protocol of Leng et al. (2024), we conduct experiments on MiniGPT-4 (Zhu et al., 2023). As shown in

Table 2, AGE consistently achieves state-of-the-art performance across all evaluation settings, surpassing the baseline by an average margin of 17.09% in Accuracy while maintaining a comparable F1 score. It is worth noting that, although OPERA (Huang et al., 2024) also intervenes on attention, it primarily penalizes certain textual attention patterns. In contrast, AGE emphasizes the dominant shifts between visual and textual attention across reasoning stages. This distinction leads to substantial performance gains: AGE outperforms OPERA by average margins of 20.09% in Accuracy and 1.25% in F1, underscoring that interventions guided by real responses and targeted to real attention behaviors yield more effective hallucination mitigation.

**Results on MME.** Fig. 4 reports results on the MME benchmark, where we follow the evaluation setup of Leng et al. (2024) and focus specifically on the hallucination subset. All experiments are conducted on LLaVA-1.5-7B (Liu et al., 2023b). Compared with VCD Leng et al. (2024), which mitigates hallucinations primarily by suppressing text bias, our AGE achieves superior performance at both the object and attribute levels. This suggests that merely reducing textual priors is insufficient; effective hallucination mitigation requires faithfully aligning text and visual attention with the dynamics observed in real responses across different reasoning stages.

Table 2: Evaluation with POPE in random, popular, and adversarial settings. We report the accuracy(%) and F1 score(%). The best results are highlighted in **bold**. [†] represents the results from Chen et al. (2024d).

| Methods | Random | | Popular | | Adversarial | | Avg. | |
|---|---|---|---|---|---|---|---|---|
| | Acc↑ | F1↑ | Acc↑ | F1↑ | Acc↑ | F1↑ | Acc↑ | F1↑ |
| Greedy [‡] | 61.00 | 71.53 | 55.33 | 68.69 | 54.00 | 67.76 | 56.77 | 69.32 |
| Beam Search [‡] | 58.00 | 69.86 | 50.33 | 66.21 | 52.00 | 66.97 | 53.44 | 67.68 |
| OPERA (Huang et al., 2024) [‡] | 57.66 | 69.97 | 51.00 | 66.82 | 52.67 | 67.58 | 53.77 | 68.12 |
| VCD (Leng et al., 2024) [‡] | 60.33 | 65.71 | 57.33 | 65.21 | 53.67 | 62.13 | 57.11 | 64.35 |
| Ours | **77.32** | **71.58** | **74.03** | **68.73** | **70.23** | **67.81** | **73.86** | **69.37** |

Table 3: Ablation study of AGE on CHAIR$_S$ ($C_S$), CHAIR$_I$ ($C_I$), and BLEU. SMA: self-multiplicative amplification of attention; AGE$_T$: text attention intervention; AGE$_I$: image attention intervention. $Max\ new\ tokens$ is 128. Best results are in **bold**.

| SMA | AGE$_T$ | AGE$_I$ | $C_S \downarrow$ | $C_I \downarrow$ | BLEU↑ |
|---|---|---|---|---|---|
| | | | 53.4 | 14.2 | 10.5 |
| ✓ | | | 43.1 | 13.1 | 10.1 |
| | ✓ | | 50.4 | 14.9 | 10.4 |
| | | ✓ | 35.4 | 10.9 | 10.4 |
| | ✓ | ✓ | **31.8** | **10.0** | **10.5** |

## 5.2 FURTHER ANALYSIS

**Contribution of Each Intervention.** We conduct ablation experiments on the COCO dataset with LLaVA-1.5. By incrementally enabling individual intervention and reporting CHAIR and BLEU scores, we examine how each intervention affects model performance. To further test robustness under different output settings, we increase the $max\ new\ token$ to 128, double the 64-token setting in Table 1, thereby validating the method's effectiveness across varying generation lengths. As shown in Table 3, SMA refers to visual attention intervention via self-multiplicative augmentation, a straightforward strategy that directly amplifies the original attention values. AGE$_T$ denotes text attention intervention through direct amplification of attention values, and AGE$_I$ represents our directional vector–based visual attention intervention. Relative to the baseline, SMA improves CHAIR$_S$ by 10.3%, showing that even simple amplification of real attention behaviors can bring notable gains. More importantly, our proposed AGE$_I$ achieves a further boost, raising CHAIR$_S$ from 10.3% to 18.0%, which highlights that precise, vector-guided interventions are substantially more effective than coarse, direction-agnostic scaling. In addition, applying AGE$_T$ alone yields a 3.0% improvement, indicating that imitating text attention patterns guided by real responses is likewise effective. Finally, combining AGE$_I$ and AGE$_T$ yields the best overall performance, indicating that the two approaches offer complementary benefits. These results validate not only the effectiveness of each intervention in isolation but also the synergistic improvements achieved when applied together. Please refer to the Appendix A.5 for more results about the different intervention layers.

**Ablation Study on Sample Size $M$.** In Sec. 4.1, we generate the target directional vector $\delta$ from $M$ sampled instances. Here, we analyze how different values of $M$ affect CHAIR and BLEU on LLaVA-1.5 with COCO, as shown in Fig. 5. Regardless of the value of $M$ (from 5 to 200), AGE consistently outperforms the baseline on both CHAIR and BLEU, indicating that the target directional vector $\delta$ can be reliably estimated with only a small number of samples, without requiring large-scale aggregation. This confirms that the performance gains of AGE do not stem from introducing additional samples. Moreover, as $M$ increases, CHAIR scores show a slight increase, suggesting that excessive samples introduce noise and reduce the accuracy of $\delta$. Balancing performance and computational efficiency, we set $M = 10$ in our experiments.

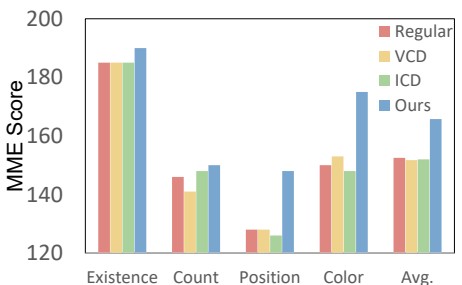

Figure 4: Evaluation on MME benchmark across the 'Existence', 'Count', 'Position', 'Color', and 'Avg.' settings. Best results are in **bold**.

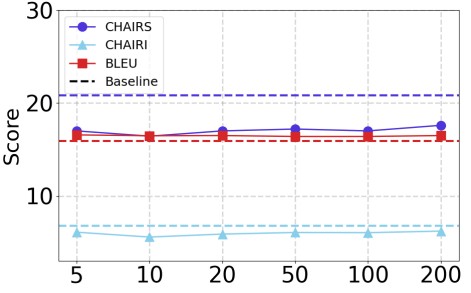

Figure 5: Impact of varying the sample size $M$ for generating $\delta$ on model performance, evaluated with CHAIR (%) and BLEU (%).

Table 4: Comparison of CHAIR (%), and BLEU (%) with different variants of AGE. $\text{AGE}_D$: $\delta$ generated from the corresponding intervention layer. $\text{AGE}_M$: Intervention in all Middle stage layers. $\text{AGE}_L$: Intervention in all Late stage layers. The best results are highlighted in **bold** while the second-best results are marked with underline.

| Method | $C_S \downarrow$ | $C_I \downarrow$ | BLEU↑ |
|---|---|---|---|
| Baseline | 20.80 | 6.77 | 15.93 |
| $\text{AGE}_D$ | 19.40 | 5.86 | 16.69 |
| $\text{AGE}_M$ | 28.40 | 8.64 | 16.46 |
| $\text{AGE}_L$ | 19.20 | 6.22 | **16.73** |
| AGE | **16.43** | **5.58** | 16.48 |

**Variants of AGE.** To further investigate AGE, we design several variants based on LLaVA-1.5, as shown in Tab. 4. Specifically, $\text{AGE}_D$ denotes using $\delta$ generated from the corresponding intervention layer, $\text{AGE}_M$ applies interventions to all Middle-stage layers, and $\text{AGE}_L$ intervenes in all Late-stage layers. $\text{AGE}_D$ outperforms the baseline, confirming that computing $\delta$ within the corresponding layer is effective. However, its performance is still weaker than that of AGE obtained from the final output layer, suggesting that the attention state closer to the output carries richer visual grounding information. In contrast, $\text{AGE}_M$ underperforms the baseline, indicating that coarse, uniformly enhancing textual attention across all Middle layers degrades model quality. We attribute this to the fact that attention disparities between real and hallucinated responses evolve dynamically across layers;

thus, indiscriminate interventions can distort attention patterns instead of aligning them. A similar trend is observed with $\text{AGE}_L$: although applying interventions to all Late-stage layers yields slight improvements over the baseline, it remains inferior to the selective interventions of AGE. Overall, these findings underscore that targeted, fine-grained interventions, not blanket modifications, are key to effective hallucination mitigation.

**Visualization of Responses.** To more intuitively demonstrate the effectiveness of our AGE, we conduct visualization experiments on LLaVA-1.5 with the query "Please describe this image in detail." As shown in Fig. 6, the baseline model hallucinates objects (red), such as "a dining table" which is absent from the image. In contrast, responses generated with AGE are more faithful and precise (green), $e.g.$, correctly noting that "broccoli occupies most of the bowl." These results indicate that AGE, by imitating truth-grounded attention behaviors, substantially enhances both the factual accuracy and descriptive richness of generated captions. Furthermore, we visualize the cross-modal attention difference ($\bar{s}^{(l)}_{(\text{real/hall,vision})} - \bar{s}^{(l)}_{(\text{real/hall,text})}$) across layers for real (blue) and hallucinated (red) tokens, and compare them with responses produced after applying AGE (green). The visualization reveals clear disparities between real and hallucinated tokens in baseline LLaVA, whereas AGE narrows this gap, aligning generated responses more closely with real, hallucination-free attention patterns. This alignment not only validates our analysis in Sec. 3.3 but also demonstrates that imitating real-response attention enables more faithful multimodal reasoning and significantly improves response quality. Additional visualizations for MiniGPT and mPLUG-Owl are provided in the Appendix A.9.

## 6  CONCLUSION

In this paper, we introduce AGE, a novel attention intervention approach designed to mitigate hallucination by imitating the attention behavior of real responses. In this way, we can capture the fine-grained dynamics of multimodal reasoning and improving response quality with less hallucanations. Extensive experiments across three benchmarks and three LVLMs validate the effectiveness

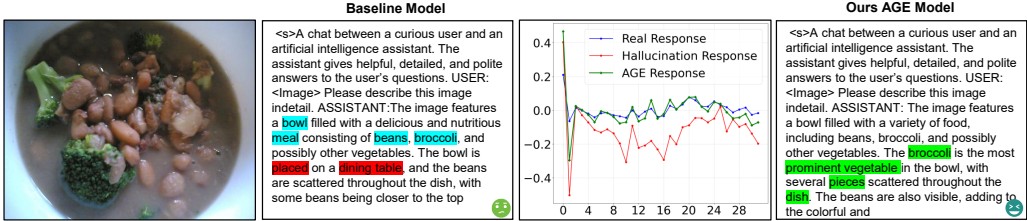

Figure 6: Visualization of responses and cross-modality attention difference for the baseline model LLaVA-1.5 versus our AGE. Red and blue text/line highlights hallucinated and real tokens generated by the baseline, respectively. while green text/line marks the calibrated tokens produced by AGE.

of our approach. We hope this work provides new insight into the internal dynamics of multimodal reasoning, offering a principled path toward building more reliable LVLMs.

## 7 ETHICS STATEMENT

This research was conducted in accordance with established ethical standards for scientific inquiry. We carefully considered issues related to human subject involvement, dataset usage and release, potential societal harms, research methodologies and applications, conflicts of interest, fairness and bias, privacy and security, legal compliance, and overall research integrity (e.g., IRB approvals and ethical documentation). Our study does not involve human participants or personally identifiable information, and thus no Institutional Review Board (IRB) approval was required. All datasets employed are publicly available and distributed under appropriate licenses. Potential risks—including bias, fairness, privacy, and unintended misuse—were critically assessed, and steps were taken to minimize such concerns. We affirm that this work adheres to principles of research integrity, ensuring accuracy, transparency, and reproducibility.

## 8 REPRODUCIBILITY STATEMENT

We have taken extensive measures to facilitate the reproducibility of our findings. The main paper provides detailed descriptions of the model architecture and methodology, while the appendix further documents implementation specifics and hyperparameter settings. All datasets used in our experiments are publicly available. To ensure faithful replication, we release our source code, configuration files, execution environment, and preprocessing scripts in an anonymous repository, enabling researchers to reliably reproduce our experiments.

## ACKNOWLEDGMENTS

The authors would like to thank the anonymous referees for their valuable comments. In this work, Hairui Ren, Zixuan Wang and Dandan Guo are supported by the National Natural Science Foundation of China (No. 62306125). Yi Chang is supported by the National Key R&D Program of China under Grant (No. 2023YFF0905400), the National Natural Science Foundation of China (No. U2341229). This work was carried out during Dandan Guo's visit to King Abdullah University of Science and Technology (KAUST).

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

# A APPENDIX

## A.1 THE USE OF LLM

In this work, we utilize Large Language Models to perform grammatical verification.

## A.2 ALGORITHM

---

**Algorithm 1** Workflow about our AGE method for hallucination mitigation

---

**Require:** Intervention Image Attention Layer Set $L_{\text{image}}$, Intervention Text Attention Layer Set $L_{\text{text}}$, Hallucination samples $M$

1: Construct a hallucinated tokens set $I_{\text{hall}}^{(i)}$

2: Construct a real tokens set $I_{\text{real}}^{(i)}$

3: Compute hallucinated average image attention vector $\overline{\mathbf{a}}_{(\text{hall,vision})}^{i} = \frac{1}{|I_{\text{hall}}^{(i)}|} \sum_{k \in I_{\text{hall}}^{(i)}} \mathbf{a}_{\text{vision}}^{(L,k)}$

4: Compute real average image attention vector $\overline{\mathbf{a}}_{(\text{real,vision})}^{i} = \frac{1}{|I_{\text{real}}^{(i)}|} \sum_{k \in I_{\text{real}}^{(i)}} \mathbf{a}_{\text{vision}}^{(L,k)}$

5: Compute the sample weight $w_i = \begin{cases} (C_I^i/\tau)^p, & \text{if } C_I^i < \tau, \\ \exp\left(-q \times (C_I^i - \tau)\right), & \text{if } C_I^i \geq \tau, \end{cases}$

6: Generate target directional vector $\boldsymbol{\delta} = \frac{1}{M} \sum_{i=1}^{M} w_i \cdot (\overline{\mathbf{a}}_{(\text{real,vision})}^{i} - \overline{\mathbf{a}}_{(\text{hall,vision})}^{i})$

7: **for** each decoding step $k$ **do**

8:     **for** each decoding layer $l$ **do**

9:         **if** layer $l$ in $L_{\text{image}}$ **then**

10:             Imitating Image Attention $\hat{\mathbf{a}}_{\text{vision}}^{l} = \mathbf{a}_{\text{vision}}^{l} + \lambda_v \times \boldsymbol{\delta}$

11:         **end if**

12:         **if** layer $l$ in $L_{\text{text}}$ **then**

13:             Imitating Text Attention $\hat{\mathbf{a}}_{\text{text}}^{l} = \mathbf{a}_{\text{text}}^{l} + \lambda_t \times \mathbf{a}_{\text{text}}^{l}$

14:         **end if**

15:         Compute next-layer hidden state $\mathbf{h}_k^{(l+1)} = \mathbf{h}_k^{(l)} +$ AttentionSubLayer$(\hat{\mathbf{a}}_{\text{vision}}^{(l,k)}, \mathbf{V}_{\text{vision}}^{(l)}, \hat{\mathbf{a}}_{\text{text}}^{(l,k)}, \mathbf{V}_{\text{text}}^{(l,k)})$

16:     **end for**

17: **end for**

---

## A.3 ADAPTIVE WEIGHT STRATEGY

In sec. 4.1, we calculate the target directional vector $\boldsymbol{\delta}$ with $M$ samples. However, assigning equal weights overlooks that different samples, due to their varying proportions of hallucinated content, contribute unequally to the corrective signal. To address this, we introduce an Adaptive Weight Strategy using sample-specific weights $w_i$ based on the CHAIR instance-level score ($C_I^i$). The key intuition is that samples with a balanced ratio of real-to-hallucinated tokens ($C_I^i \approx \tau$) contain richer comparative signals, making them more informative for constructing the difference vector. Conversely, samples heavily dominated by hallucinations ($C_I^i \gg \tau$) can inject misleading bias into $\boldsymbol{\delta}$, and should therefore be penalized. A piecewise weighting function is adopted to reflect this asymmetric importance: one branch smoothly increases the weight as $C_I^i$ approaches $\tau$ from below, while the other rapidly decreases it when $C_I^i$ exceeds the threshold, ensuring that excessively noisy samples do not dominate the correction: We define $w_i$ as:

$$w_i = \begin{cases} (C_I^i/\tau)^p, & \text{if } C_I^i < \tau, \\ \exp\left(-q \times (C_I^i - \tau)\right), & \text{if } C_I^i \geq \tau, \end{cases}$$

where $\tau$ denotes the balance threshold. The exponent $p$ governs the amplification rate for balanced samples, and $q$ controls the suppression strength for hallucination-dominated ones. In our implementation, $p = 1$ and $q = 30$, determined empirically on a small held-out validation set.

Table 5: Comparison of CHAIR and BLEU with different image attention intervention layers. For LLaVA, we do not intervene with image attention in the Middle Stage; only intervene with text attention is needed, while other models intervene in all Stages. The best results are in **bold**.

| Method | MiniGPT-4-7B | | | LLaVA-1.5-7B | | | mPLUG-Owl2-7B | | |
|---|---|---|---|---|---|---|---|---|---|
| | $C_S \downarrow$ | $C_I \downarrow$ | BLEU↑ | $C_S \downarrow$ | $C_I \downarrow$ | BLEU↑ | $C_S \downarrow$ | $C_I \downarrow$ | BLEU↑ |
| baseline | 30.87 | 12.33 | 14.33 | 20.80 | 6.77 | 15.93 | 23.20 | 8.33 | 15.37 |
| 1,2 | **8.80** | **4.73** | 12.38 | 0.00 | 0.00 | 1.03 | 22.20 | 8.55 | 16.12 |
| 3,4 | 2.20 | 3.31 | 4.93 | 19.00 | 6.05 | 16.60 | 20.60 | 7.84 | 16.29 |
| 5,6 | 2.00 | 15.85 | 3.68 | 0.20 | 1.03 | 2.96 | 20.40 | 8.15 | 16.25 |
| 7,8 | 14.60 | 9.71 | 14.82 | 20.00 | 6.63 | **16.70** | 19.20 | 7.70 | 16.30 |
| 9,10 | 13.80 | 9.44 | 14.60 | 19.00 | 6.36 | 16.50 | 21.80 | 8.20 | **16.34** |
| 11,12 | 0.20 | 2.12 | 0.56 | 17.80 | 5.87 | 16.17 | 23.20 | 8.21 | 16.23 |
| 13,14 | 18.80 | 7.61 | 15.64 | 17.80 | 5.96 | 16.43 | 22.60 | 8.66 | 16.38 |
| 15,16 | 11.20 | 4.00 | 15.37 | 19.40 | 6.66 | 16.52 | 23.00 | 8.50 | 16.29 |
| 17,18 | 15.60 | 5.57 | **16.99** | - | - | - | 22.20 | 8.06 | 16.24 |
| 19,20 | 15.80 | 6.12 | 15.78 | - | - | - | 20.80 | 7.56 | 16.30 |
| 21,22 | 18.20 | 7.52 | 15.54 | - | - | - | 21.40 | 8.02 | 16.26 |
| 23,24 | 17.20 | 5.78 | 16.12 | - | - | - | 21.20 | 7.97 | 16.29 |
| 25,26 | 17.00 | 6.13 | 15.68 | - | - | - | 22.00 | 8.18 | 16.26 |
| 27,28 | 15.91 | 6.19 | 15.86 | 18.00 | 6.10 | 16.60 | 21.40 | 8.03 | 16.26 |
| 29,30 | 19.00 | 6.76 | 16.05 | 17.40 | 5.75 | 16.64 | 21.40 | 7.99 | 16.32 |
| Ours (30,31) | 15.62 | 6.00 | 15.79 | **16.43** | **5.58** | 16.48 | **19.40** | **7.47** | 16.21 |

## A.4 Evaluation Metrics

The CHAIR$_S$ and CHAIR$_I$ scores are computed by comparing the model-generated answers with the ground truth object annotations as :

$$CHAIR_I = \frac{\|\{hallucinated\ objects\}\|}{\|\{all\ mentationde\ objects\}\|}, CHAIR_S = \frac{\|\{captions\ with\ hallucinated\ objects\}\|}{\|\{all\ captions\}\|}. \tag{10}$$

In addition, following the standard BLEU definition (Papineni et al., 2002), we incorporate an instance-level BLEU score to evaluate whether the generated descriptions faithfully capture the necessary visual content from the image as:

$$BLEU = \text{BP} \times \exp(\sum_{n=1}^{N} w_n log p_n) \tag{11}$$

where $p_n$ is the precision of $n$-grams between the generated and reference captions, $w_n$ is the weight assigned to $n$-grams (typically $w_n = \frac{1}{n}$), and $BP$ denotes the brevity penalty to penalize overly short predictions. We report the average BLEU score, computed as the mean of BLEU-1 through BLEU-4.

## A.5 Intervention on different layers

In our experimental setup, interventions on Late-stage image attention are applied to layers 30 and 31, while Middle-stage interventions target layer 20. This naturally raises the question: what happens if interventions are applied to other layers? To investigate, we conduct further exploration.

As shown in Tab. 5, intervening in Early-stage layers (*e.g.*, 1, 2, 5, and 6) produces extremely low $C_S$ and $C_I$ values—nearly zero—suggesting strong hallucination suppression. However, this comes at the cost of a sharp drop in BLEU, as the model fails to generate meaningful text and instead outputs repetitive or corrupted tokens (e.g., long strings of "nobody" or garbled symbols like "&&#"). We attribute this to the role of early layers, which primarily capture low-level features of images and text. Although differences in attention exist between real and hallucinated responses at these shallow layers, they do not reflect the high-level semantic inconsistencies that drive hallucinations. As a

Table 6: Comparison of CHAIR and BLEU with different text attention intervention layers in LLaVA. The best results are in **bold**.

| Layer | $C_S \downarrow$ | $C_I \downarrow$ | BLEU$\uparrow$ |
|---|---|---|---|
| Baseline | 20.8 | 6.7 | 15.9 |
| 16 | 20.6 | 7.0 | 16.0 |
| 17 | 22.6 | 7.5 | 16.8 |
| 18 | 19.0 | 6.3 | 16.5 |
| 19 | 20.0 | 6.5 | 16.2 |
| 20 | **16.4** | **5.5** | 16.4 |
| 21 | 20.0 | 7.1 | 16.7 |
| 22 | 18.8 | 6.0 | **16.8** |
| 23 | 17.0 | 5.7 | 16.5 |
| 24 | 20.0 | 6.4 | 16.5 |
| 25 | 19.8 | 7.1 | 16.7 |
| 26 | 19.4 | 6.5 | 16.8 |

Table 7: Comparison of CHAIR and BLEU with LLaVA-1.5-13B as baseline. The best results are in **bold**.

| Method | $C_S \downarrow$ | $C_I \downarrow$ | BLEU$\uparrow$ |
|---|---|---|---|
| LLaVA-1.5-13B | 48.60 | 12.38 | 10.57 |
| AGE-13B | **45.40** | 12.55 | **10.75** |

result, interventions here disrupt the extraction of essential features, leading the model to deviate during later inference. Therefore, we avoid applying interventions in shallow layers.

In the Late stage, we compared interventions applied to layers 27–28 with those applied to layers 30–31. When targeting layers 27 and 28, $C_S$ and $C_I$ decreased by 2% and 0.67%, respectively, while BLEU improved by 0.67%, indicating that hallucinations are partially mitigated. However, the overall performance remained inferior to interventions at layers 30 and 31. We attribute this to the proximity of layers 30 and 31 to the output: the closer the intervention is to the decoding output layer, the more directly it influences the model's final predictions. Consequently, intervening at layers 30 and 31 yields the most substantial benefit, and we adopt them as the Late-stage intervention layers.

For the Middle stage, we experiment with layers 19, 20, 21, and 22, as shown in Tab. 6. All tested layers reduce hallucinations to varying degrees, and even at layer 22, BLEU reached as high as 16.87. This consistency arises because Middle-stage layers emphasize textual attention, and reinforcing it—regardless of the exact layer—improves performance. The relatively stable gains across different layers demonstrate the robustness of our approach: interventions need not be tied to a single fixed layer but rather to a stage interval, within which similar benefits can be achieved. In practice, we select the best-performing candidate (*e.g.*, layer 20) as our Middle-stage intervention layer.

Taken together, these findings highlight that interventions in shallow layers are detrimental, Middle-stage layers yield broadly consistent improvements by reinforcing textual attention, and Late-stage layers closest to the output provide the strongest influence on hallucination mitigation. Accordingly, we adopt layer 20 for Middle-stage intervention and layers 30–31 for Late-stage intervention, striking a balance between stability and effectiveness.

### A.6 INTERVENTION ON LARGE-SCALE LVLM

To evaluate the scalability of our approach, we conducted experiments on LLaVA-1.5-13B (Liu et al., 2023b). As shown in Tab. 7, AGE effectively reduces hallucinations in larger models, yielding a 3.2% decrease in $C_S$ and a 0.18% in BLEU, while $C_I$ shows only a marginal rise of 0.17%. These

results highlight that hallucinations caused by attention misalignment persist across model scales, and demonstrate the cross-scale generalization and robustness of AGE in mitigating such errors.

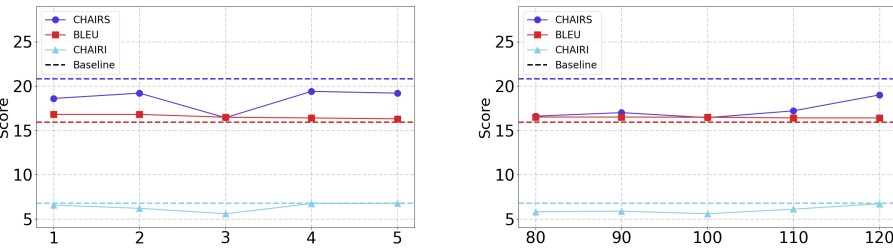

Figure 7: Comparison of CHAIR and BLEU with different values of $\lambda_t$ (left) and $\lambda_v$ (right).

## A.7 HYPERPARAMETER

To assess the sensitivity of AGE to hyperparameter choices, we use LLaVA-1.5-7B as the baseline and vary $\lambda_t$ from 1 to 5 and $\lambda_v$ from 80 to 120, reporting the corresponding CHAIR and BLEU scores. As shown in Fig. 7, performance consistently surpasses the baseline across all settings, suggesting that once the correct intervention mode is applied, hyperparameter variations have limited impact on overall gains. We therefore set $\lambda_t = 3$ and $\lambda_v = 100$ as the default configuration in our experiments.

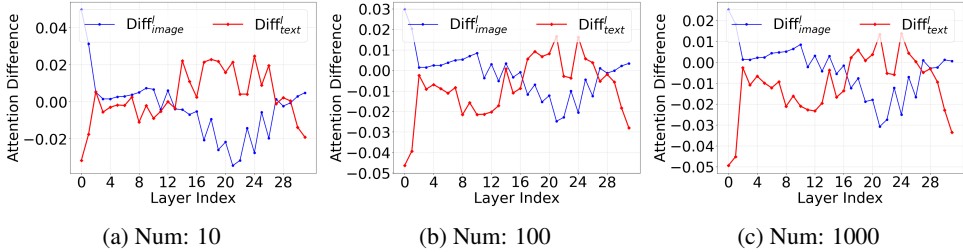

(a) Num: 10          (b) Num: 100          (c) Num: 1000

Figure 8: LLaVA-1.5 layer-wise characterization of attention disparities between real and hallucinated responses with different numbers of samples $N$.

## A.8 ANALYSIS OF MORE SAMPLES

In Sec. 3.3, we examined attention disparities by sampling $N = 100$ responses and concluded that hallucinations do not stem from a uniform lack of visual attention, but rather from the failure to reproduce nuanced, model-specific, and stage-dependent attention dynamics. A natural question arises: does this conclusion hold under different sample sizes? To test this, we compare the attention behaviors of real and hallucinated tokens in LLaVA-1.5 with $N = 10$, 100, and 1000 samples. As shown in Fig. 8, while the detailed attention curves fluctuate with sample size, the dominant modality at each reasoning stage remains consistent. This robustness further confirms the reliability and generalizability of our statistical findings.

## A.9 MORE VISUALIZATIONS

As shown in Fig. 9–12, we provide additional visual comparisons between the baseline models and our AGE on the image captioning task.

## A.10 VISUALIZATIONS OF DISTINGUISHING BETWEEN HALLUCINATION AND REAL OBJECTS

We illustrate the full procedure for distinguishing hallucinated objects from real ones in Tab. 8. Specifically, we first extract objects from the LVLM's output and from the ground-truth labels. Then, leveraging COCO's synonym vocabulary, we match the two sets to identify overlaps, thereby

separating hallucinated from real objects. The detailed steps are as follows: Following prior work (Rohrbach et al., 2018), we use the COCO ground-truth object annotations as supervision. The procedure is: (1) **Extract noun tokens.** We first identify all noun tokens in the model's generated response using a POS-tagging parser. Only nouns are considered candidate visual entities. (2) **Map each noun to object categories.** Each noun is normalized—lowercased, lemmatized, and matched to COCO object categories using a synonym and alias dictionary (the same strategy as prior CHAIR-style evaluation). This allows us to automatically determine whether a generated noun corresponds to a visually grounded category. (3) **Compare against ground-truth annotations.** If a noun (or its synonym) appears in the COCO ground-truth object list for that image, it is labeled as a real token; otherwise, it is labeled as a hallucinated token. (4) **Fully automated pipeline.** No human annotators participate in token-level labeling. The entire process—noun extraction, synonym matching, and GT comparison—is fully automated and reproducible.

## A.11 DOMAIN SHIFT GENERALIZATION

We conduct extensive domain-shift experiments across multiple LVLMs. Specifically, we directly apply the $\delta$ computed on COCO to five out-of-domain settings, including Medical (Wu et al., 2024), Video, Math, Map, and Table (Guan et al., 2024), which differ substantially from natural images. For the medical benchmark, we follow Wu et al. (2024) and report Accuracy under FAKE and SWAP settings; for the remaining domains, we report Accuracy/F1 following Guan et al. (2024). As shown in Table 9, AGE consistently improves over the baseline across all domains and architectures (*e.g.*, +3.2% in Medical for LLaVA, +3.8% in Math for MiniGPT-4, +1.6% in Table for mPLUG-Owl2). These results demonstrate that the calibrated hallucination-to-truth signal encoded by $\delta$ and the chosen stages remain effective even under severe distribution shift, indicating strong robustness and transferability rather than dataset-specific tuning. Our domain-shift experiments—covering medical images, math diagrams, tables, and video frames—confirm that AGE generalizes well even in tasks without object-level supervision.

## A.12 LATENCY AND THROUGHPUT

To assess the computational efficiency of AGE, we report both latency and throughput across three different LVLM architectures (Table 10). The results show that AGE introduces only negligible latency overhead while maintaining high throughput on all tested models.

## A.13 A DYNAMIC VERSION OF AGE

To completely eliminate dependencies on manual layer selection and hyperparameter tuning, we further develop Dynamic AGE (see Appendix A.13). This variant replaces fixed settings with an adaptive mechanism that monitors the live attention gap at every decoding step. The procedure consists of three phases:

Phase 1: Constructing Reference Profiles (Offline). We first aggregate attention statistics from the $M$ auxiliary samples to build a "Hallucination Reference Profile," representing the typical attention state of hallucinated tokens: * **Visual Reference Vector $(a^*_{(hall)})$:** The weighted average of visual attention vectors from hallucinated tokens:

$$a^*_{(hall,vision)} = M \sum_{i=1}^{M} w_i \cdot (\overline{a}^i_{(hall,vision)})$$

Textual Reference Sum $(s^*_{(hall)})$: The weighted average of attention sums allocated to text tokens in hallucinated responses:

$$s^*_{(hall,text)} = M \sum_{i=1}^{M} w_i \cdot \overline{s}^i_{(hall,text)}$$

Phase 2: Calculating Dynamic Gaps (Online). During inference, for each layer $l$ and decoding step $k$, we calculate how much the current attention state deviates from the hallucination profile: Visual Instantaneous Deviation $(\delta_d)$: We define $\delta_d$ as the vector difference between the current

visual attention $a_{vision}^{l,k}$ and the hallucination reference. This captures the \*live\* drift of the current token:

$$\delta_d^{(l,k)} = a_{vision}^{l,k} - a_{(hall,vision)}^*$$

Dynamic Visual Scaling ($g_v$): We determine the intervention strength by normalizing the magnitude (RMS) of this instantaneous deviation ($\delta_d$) against the global calibration vector ($\delta = \frac{1}{M} \sum_{i=1}^{M} w_i \cdot (\overline{\mathbf{a}}_{(real,vision)}^i - \overline{\mathbf{a}}_{(hall,vision)}^i)$, derived in Eq. 6). This ratio quantifies how "strong" the current deviation is relative to the expected correction direction:

$$g_v^l = \frac{RMS(\delta_d^{(l,k)})}{RMS(\delta)}$$

Dynamic Textual Scaling ($g_t$): Similarly, we measure the ratio of the current text attention sum to the hallucination reference sum:

$$g_t^l = \frac{s_{text}^{(l,k)}}{s_{(hall,text)}^*}$$

Phase 3: Adaptive Calibration. Crucially, intervention is triggered only when the dynamic gap implies a significant mismatch with truthful patterns (i.e., exceeding a statistical threshold $\tau$). The model automatically self-calibrates using the computed dynamic strengths:

Visual Calibration: If the dynamic gap $g_v^l$ exceeds $\tau_v$, we inject the global correction vector $\delta$ scaled by the live strength $g_v^l$:

$$\hat{a}_{vision}^l = a_{vision}^l + g_v^l \times \delta, \quad \text{if } g_v^l > \tau_v$$

Textual Calibration: If the textual attention ratio $g_t^l$ exceeds $\tau_t$ (indicating a need for stronger context grounding), we perform self-multiplicative enhancement:

$$\hat{a}_{text}^l = a_{text}^l + g_t^l \times a_{text}^l, \quad \text{if } g_t^l > \tau_t$$

Result: This mechanism enables the model to automatically determine both which layers to intervene in and how much modulation strength to apply at each step (via $g_v^l, g_t^l$), without any manual tuning. Following the logic used for $\lambda_v$ and $\lambda_t$, we initialize their base values to 100 and 1, respectively. The terms $\tau_v$ and $\tau_t$ capture the instantaneous discrepancy between the current token's attention and the hallucination pattern, and simultaneously act as dynamic weights that guide the calibration strength. We evaluate this dynamic variant of AGE in Table 11. While AGE$_{Dynamic}$ offers strong flexibility in both layer choice and weighting, its slightly higher $C_S$ and $C_I$ scores—relative to manually tuned AGE—suggest a modest performance trade-off. We attribute this to the more aggressive and unconstrained adjustments occasionally interfering with the preservation of low-level visual or textual features, reducing the net benefit of semantic-level calibration. Nevertheless, AGE$_{Dynamic}$ remains substantially better than the baseline across all metrics, demonstrating that the core mechanism of AGE is inherently robust, adaptive, and transferable, even without manually specified parameters.

### A.14 An Online version of AGE

Fully online estimation is promising. However, without ground-truth supervision during inference, online estimation is challenging. Since the model cannot reliably distinguish real tokens from hallucinatory ones, making it impossible to compute an accurate $\boldsymbol{\delta}$ vector as we do offline. In our original submission, we had devised a more direct online calibration mechanism that avoids using $\boldsymbol{\delta}$. Instead of relying on real-hallucination discrepancies, the online variant applies self-multiplicative enhancement to both image and text attention, allowing the model to internally amplify reliable signals. As shown in Table 12, this online version does not match the performance of the offline $\boldsymbol{\delta}$-guided method, but it still achieves consistent and notable gains over the baseline, demonstrating that effective hallucination mitigation is possible even without offline supervision. These results further support the core motivation of our work: The model benefits from imitating truth-aligned attention behaviors, and this principle remains effective even when the calibration is performed online without explicit $\boldsymbol{\delta}$.

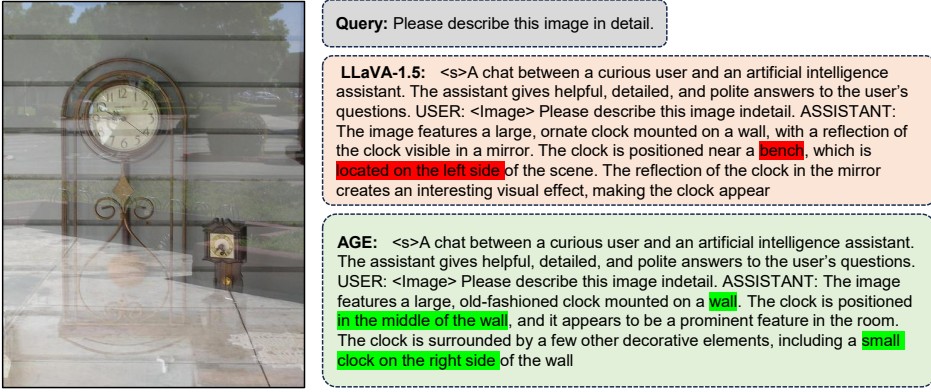

Figure 9: Generated image description examples.

## A.15 COMPARISON WITH MORE METHODS

Some studies (VTI (Liu et al., 2024d), Nullu (Yang et al., 2025a), AGLA (An et al., 2024), Deco (Wang et al., 2024a), and DeGF (Zhang et al., 2025)) have similarities with our AGE, so we further compare and analyze AGE with these studies. VTI (Liu et al., 2024d), which mitigates hallucinations by stabilizing visual representations through noise-perturbed image augmentation; Nullu (Yang et al., 2025a), which contrasts truthful and hallucinated responses at the sample level and intervenes in weight space; DeGF (Zhang et al., 2025), which leverages text-to-image generative feedback to guide correction; AGLA (An et al., 2024), which assembles global and local attention to better capture object semantics; DeCo (Wang et al., 2024a), which dynamically corrects decoding by aligning confidence patterns across layers.

We directly compare AGE with AGLA, DeCo, and DeGF, and also integrate AGE into each system. As shown in Table 13, AGE alone achieves the best or comparable performance, and when combined with AGLA/DeCo/DeGF, it consistently improves them in most of settings. This confirms that AGE introduces non-overlapping, complementary gains, supporting that our layer-wise, truth-guided attention imitation captures a fundamentally different source of hallucination. (2) Additionally, we originally did not compare AGE with HALC (Chen et al., 2024d) because HALC relies on an external detector (Grounding DINO), which introduces additional supervision beyond our method. For completeness, we now include HALC in Table 13. The results show that AGE—despite requiring no auxiliary models—still achieves SOTA performance. This further demonstrates the effectiveness and independence of our hallucination-to-real-attention calibration. (3) For qualitative analysis, we include two visualization studies in Fig. 14 and Fig. 15. As the results show, DeGF, DeCo, and AGLA all hallucinate the same nonexistent objects—such as the "chair" in Fig. 14 and the "wine glasses" and "cups" in Fig. 15. In contrast, AGE produces descriptions that are both more accurate and more detailed, offering a clearer and more intuitive demonstration of its effectiveness in reducing hallucinations.

In summary, AGE differs from prior approaches by **(1)** identifying the token-level, stage-specific attention patterns that characterize truthful responses, **(2)** revealing model-dependent text-image attention requirements, and **(3)** intervening directly along these dynamics. These insights form the core conceptual novelty of our work.

## A.16 FAILURE CASE ANALYZE

We further examine the failure cases-images that still exhibit hallucinations after applying AGE, and observe a consistent pattern: as shown in Fig. 13, AGE's calibration ability diminishes in scenes containing a large number of distinct small objects. In such complex settings, the abundance of visual instances increases the difficulty of accurate grounding, leading the model to continue producing hallucinations. We hypothesize that these cases require more fine-grained extraction and reinforcement of visual cues, which offers a clear direction for future improvements.

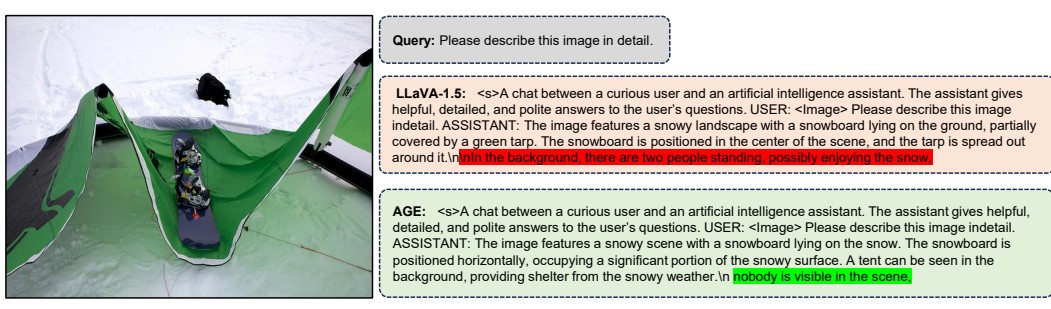

Figure 10: Generated image description examples.

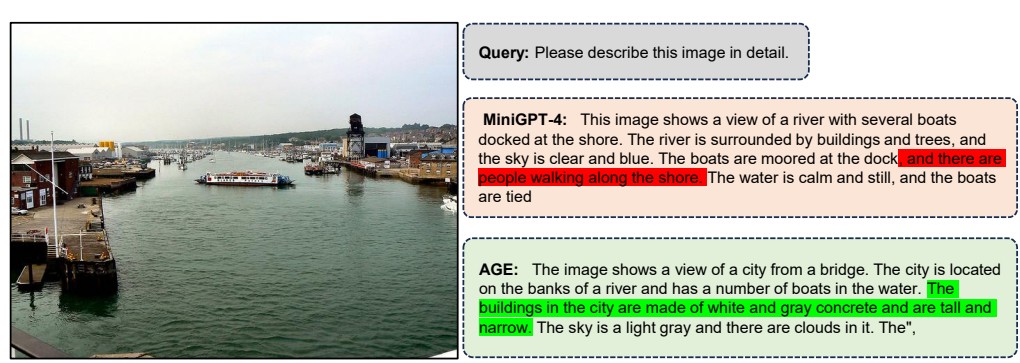

Figure 11: Generated image description examples.

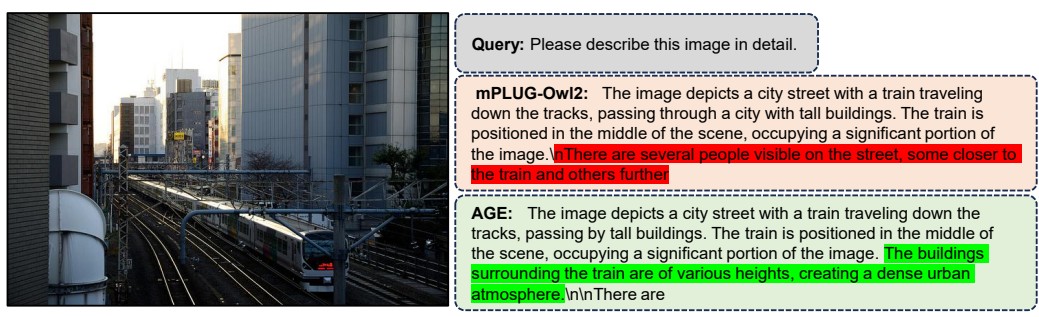

Figure 12: Generated image description examples.

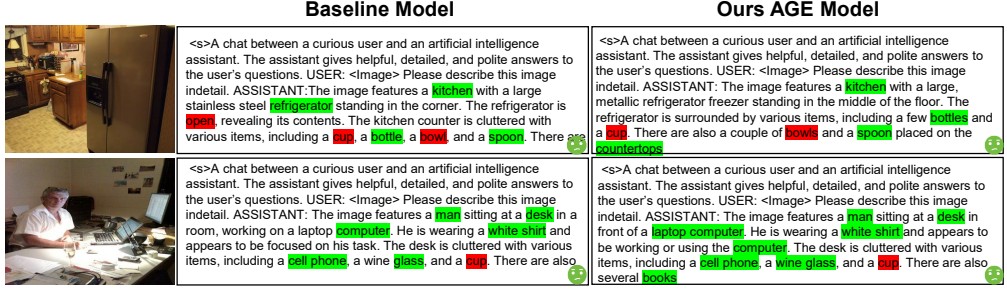

Figure 13: An example of the failure case in AGE. Real token is marked green, hallucination is marked red.

Table 8: An example to demonstrate how to distinguish between hallucinatory objects and real objects.

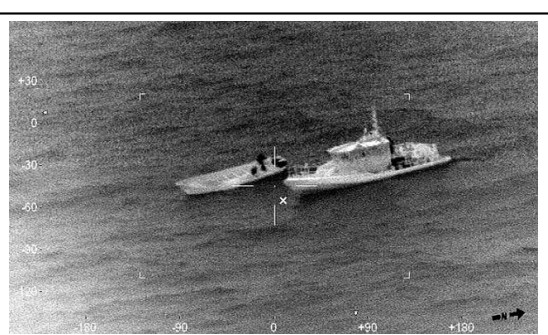

| Instruction | Please describe this image in detail |
|---|---|
| LLaVA Output | The image features two boats floating on a body of water, possibly the ocean. One boat is larger and positioned towards the left side of the scene, while the other boat is smaller and located closer to the center. The smaller boat appears to be a speedboat, while the larger boat is a yacht. There are three people visible in the scene, with one person standing on the larger yacht and two others on the smaller speedboat. The people seem to be enjoying their time on the water, possibly engaging in recreational activities or simply enjoying the view. |
| Generated Objects | boats, speedboat, yacht, people, person |
| Ground Truth Caption | an old photo os a small boat near a bigger boat in a large body of water; Two boats being spied on from a distance; One large and one small boat floating in the ocean; One boat sailing next to one canoe in a body of water; two white boats next to each other in the water; |
| Ground Truth Objects | boat |
| Hallucination Objects | people, person |
| Real Objects | boats, speedboat, yacht |

**[Prompt]**

Please describe this image in detail.

**[Image]**

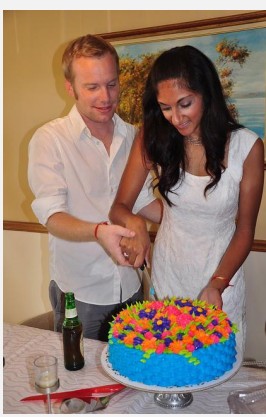

**[DeGF]**

The image features a man and a woman standing next to a colorful cake, both smiling and preparing to cut it together. The cake is decorated with flowers and is placed on a dining table. The couple appears to be enjoying the moment and sharing a special occasion.\n\nThere are a few bottles on the table, possibly containing beverages for the celebration. A chair can be seen in the background, and a handbag is placed nearby. The scene captures the joy and excitement of the couple as they cut into the cake together.

**[Deco]**

The image depicts a man and woman standing together, both holding onto a knife as they cut into a birthday cake together. They appear to be celebrating a birthday party or similar occasion. The cake itself is colorful and decorated with flowers, making it look festive and delicious.\n\nThere are several bottles placed around the scene, likely containing drinks for the party guests. A chair can also be seen in the background, possibly for guests to sit down and enjoy their refreshments.

**[AGLA]**

The image depicts a man and a woman standing next to each other, both holding a knife together and cutting a cake. They are smiling and enjoying the moment. The cake is placed on a dining table, which occupies a significant portion of the scene.\n\nThere are a few other items in the scene, such as a bottle located near the left side of the table, and a chair positioned at the far left side of the image. The man and woman appear to be the main focus of the scene, as they share the experience of cutting the cake together.

**[AGE]**

The image depicts a man and a woman standing next to each other, both holding a knife and cutting a large, multi-colored cake. The cake is placed on a dining table, and there is a bottle of beer on the table as well. The couple appears to be celebrating a special occasion, possibly their wedding anniversary, as they share the joyful moment together.

Figure 14: An example of generated captions by different decoding methods. Hallucinated content is marked in red.

**[Prompt]**

Please describe this image in detail.

**[Image]**

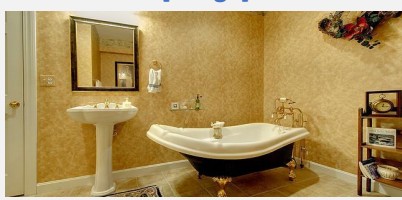

**[DeGF]**

The image features a luxurious bathroom with a large, black bathtub sitting on a tile floor. The bathtub is positioned in the center of the room, surrounded by walls that are painted beige. A mirror is placed on the left side of the room, and a sink can be seen on the left side as well.\n\nThe bathroom is well-stocked with various toiletries, including multiple bottles and cups placed around the room. There are also a couple of wine glasses and a bowl, adding to the elegant atmosphere of the space. The bathroom appears to be a part of a hotel suite, providing a comfortable and relaxing environment for guests.

**[Deco]**

The image depicts a large bathtub placed against a wall in a bathroom. The bathtub occupies most of the space in the room, extending from one end of the room to the other. Next to the bathtub, there is a sink with a mirror above it. The sink area also contains a bottle placed on top of it.\n\nIn addition to these items, there are two bottles placed near the bathtub, one on each side of it. Furthermore, there are two cups placed close to each other near the bathtub as well. The bathroom appears to be well-equipped with bathroom essentials.

**[AGLA]**

The image features a large, white claw-foot bathtub situated in a bathroom. The tub is placed on a tile floor, and it appears to be empty. The bathroom is well-equipped with a sink, a mirror, and a towel rack. A bottle can be seen on the sink counter, and a wine glass is placed nearby.\n\nIn addition to the main elements, there is a vase and a cup on the counter, adding to the overall decoration of the bathroom. The bathroom also has a toilet, which is located in the background.

**[AGE]**

The image features a spacious bathroom with a large, claw-foot tub positioned in the center of the room. The tub is surrounded by a tiled floor, and it appears to be the focal point of the bathroom. The walls are adorned with a beige wallpaper, and the room is illuminated by a dim light.\n\nIn addition to the tub, there are several other items in the bathroom. A sink is located on the left side of the room, and a mirror is placed above it. There are also two bottles, one placed on the left side of the tub and the other on the right side. A towel is hanging on the wall, and a vase can be seen on the right side of the room.

Figure 15: An example of generated captions by different decoding methods. Hallucinated content is marked in red.

Table 9: Source $\delta$ is generated by COCO. Comparison with different domain benchmarks: Medical (Wu et al., 2024), Video (Guan et al., 2024), Math (Guan et al., 2024), Map (Guan et al., 2024), and Table (Guan et al., 2024). FAKE question: Fake or nonsensical questions are used to examine model's ability to detect incoherent questions. Image SWAP: In this scenario, we swap the images with unrelated ones to evaluate the model's ability to detect mismatches between the image content and the question. The best results are highlighted in **bold**.

| Methods | Medical | | Video | |
|---|---|---|---|---|
| | FAKE ↑ | NONE ↑ | Acc ↑ | F1 ↑ |
| LLaVA | 20.9 | 5.8 | 57.7 | 53.2 |
| Ours | **24.5** | **8.6** | **58.7** | **54.0** |
| | Math | | Map | |
| | Acc ↑ | F1 ↑ | Acc ↑ | F1 ↑ |
| MiNiGT4 | 66.6 | 40.0 | 25.0 | 24.9 |
| Ours | **68.2** | **46.0** | **26.9** | **26.8** |
| | Table | | Map | |
| | Acc ↑ | F1 ↑ | Acc ↑ | F1 ↑ |
| MPLUG-Owl2 | 72.7 | 52.2 | 36.5 | 41.4 |
| Ours | **74.5** | **53.5** | **41.3** | **44.8** |

Table 10: Comparison of latency (seconds/tokens) and throughput (tokens/seconds) with different LVLMs.

| Method | MiniGPT-4-7B | | LLaVA-1.5-7B | | mPLUG-Owl2-7B | |
|---|---|---|---|---|---|---|
| | $latency(s/t)\downarrow$ | $throughput(t/s)\uparrow$ | $latency(s/t)\downarrow$ | $throughput(t/s)\uparrow$ | $latency(s/t)\downarrow$ | $throughput(t/s)\uparrow$ |
| Baseline | 7.1e-5 | 14104.4 | 1.7e-4 | 5700.8 | 2.1e-4 | 4635.4 |
| Ours | 6.8e-5 | 14646.9 | 1.8e-4 | 5418.6 | 1.9e-4 | 5234.2 |

Table 11: Comparison between the AGE-Dynamic variants on LLaVA-1.5. The best results are shown in **bold**.

| Method | $C_S\downarrow$ | $C_I\downarrow$ | BLEU↑ |
|---|---|---|---|
| Baseline | 20.80 | 6.77 | 15.93 |
| $\text{AGE}_{Dynamic}$ | 17.77 | 5.89 | **16.80** |
| AGE | **16.43** | **5.58** | 16.48 |

Table 12: Comparison between the AGE-Online variants on LLaVA-1.5. The best results are shown in **bold**.

| Method | $C_S\downarrow$ | $C_I\downarrow$ | BLEU↑ |
|---|---|---|---|
| Baseline | 20.80 | 6.77 | 15.93 |
| $\text{AGE}_{Online}$ | 16.93 | 6.05 | 16.13 |
| AGE | **16.43** | **5.58** | 16.48 |

Table 13: Hallucination rates (%) are reported using CHAIR$_S$ ($C_S$), CHAIR$_I$ ($C_I$), and BLEU (%) on COCO image captioning tasks, where lower CHAIR and higher BLEU are better. Intervention Target and Intervention Modality are reported with each method. The $max\ new\ token$ is set to 512. The best results are highlighted in **bold**. [†] represents the results reported from the corresponding original paper. [‡] represents the results reported from the corresponding official code.

| Method | Intervention Target | Intervention Modality | $C_S \downarrow$ | $C_I \downarrow$ | Recall↑ |
|---|---|---|---|---|---|
| Baseline | - | - | 51.0 | 15.2 | 75.2 |
| VTI (Liu et al., 2024d) [†] | Hidden State | Text & Image | 35.8 | 11.1 | 76.8 |
| Nullu (Yang et al., 2025a) [‡] | Weights | Image | 54.6 | 14.1 | **81.3** |
| HALC (Chen et al., 2024d) [‡] | Logits | Image | 41.8 | 12.2 | 80.3 |
| Ours | Attention | Text & Image | **35.6** | **10.8** | 77.2 |
| DeGF (Zhang et al., 2025) [‡] | Logits | Image | 46.0 | 14.2 | 78.5 |
| Ours + DeGF | Attention & Logits | Text & Image | **45.8** | **14.0** | **79.3** |
| AGLA (An et al., 2024) [†] | Logits | Image | 43.0 | 14.1 | 78.9 |
| Ours + AGLA | Attention & Logits | Text & Image | **40.1** | **13.0** | **81.2** |
| Deco (Wang et al., 2024a) [†] | Hidden State | Image | 37.8 | 11.1 | **77.6** |
| Ours + Deco | Attention & Hidden State | Text & Image | **32.6** | **10.9** | 77.5 |

