# OpenReview forum: "Imitating the Truth: Attention-aware Truth-Guided Enhancement for Hallucination Mitigation in Large Vision-Language Models"
_ICLR.cc/2026/Conference — ICLR 2026 Poster_

### Official Review · Reviewer_yxZC · 2025-10-30

**Soundness:** 3
**Presentation:** 3
**Contribution:** 3
**Rating:** 6
**Confidence:** 5

**Summary:**

This paper observes layer-wise attention disparities between real and hallucinated tokens across three LVLMs, which motivates a method for calibrating stage-specific attention behaviors using a pre-calculated directional vector on a labeled subset of data.

**Strengths:**

1. The method is clearly motivated by empirical observations, offering new insights into attention behavior in LVLMs. Comprehensive experiments demonstrate that it effectively mitigates object hallucinations and outperforms prior work.

2. The paper is well written and supported by clear figures.

**Weaknesses:**

1. My main concern is that this method relies on access to a labeled subset of data to calculate the directional vector that captures the attention correction tendency. This limits its applicability in real-world scenarios where labeled data from the same distribution are not available, and there is no validation showing whether the learned attention behavior remains valid.

2. Related to W1, it is also unclear how the stage-specific settings (line 300–302) are determined in practice, since they seem to be tuned like hyperparameters.

**Questions:**

1. In Section 3.3 and Figure 3, the statement that “the final reasoning stage (Layers 26–31) exhibits a stable and pronounced positive gap” (lines 204–205) is less convincing. The earlier layers (Layers 0–4) appear to show a more consistent and significant gap.

2. The related work section could be more up to date. most of the papers are from 2024.

Happy to raise score if my concern is addressed.

---

> ### Author Response · Authors · 2025-11-21
>
> **Response to W1 about applicability in real-world scenarios (domain shift):**
>
>  Thank you for raising this important point. We further analyze the behavior of our AGE across different architectures and under domain-shift scenarios. Please see the details in our **global response 3 about generalization AGE in domain shift benchmark**. The results demonstrate that the application of $\delta$ in AGE remains robust and generalizable across diverse models and out-of-domain settings.
>
> **Response to W2 about stage setting:**
>
> Thank you for your insightful comments. Please see the **global response 2** for the robustness of stage setting and **global response 1** about identifying stages automaticly.
>
> **Response to Q1 about early layers:**
>
> We appreciate the careful observation. Please see the **global response 2** for "Why early-layer is not recommended".
>
> **Response to Q2 about related work:**
>
> Thank you for the suggestion.
>
> We have updated the related work section to incorporate the most recent advances (2024–2025). Specifically, we now discuss:
>
> - **VTI    [3]:** mitigates hallucinations by stabilizing visual representations through noise-perturbed image augmentation.
> - **Nullu  [4]:** contrasts truthful and hallucinated responses at the sample level and intervenes in weight space.
> - **HALC [5]:** leverages distinct fine-grained optimal visual information and operates on both local and global contexts simultaneously.
> - **AGLA [6]:** assembles global and local attention to better capture object semantics.
> - **DeCo [7]:** dynamically corrects decoding by aligning confidence patterns across layers.
> - **DeGF [8]:** leverages text-to-image generative feedback to guide correction.
>
> We also include direct quantitative and qualitative comparisons with these methods in **Table 13** below and **Fig.14,15**, confirming that our method offers a distinct and orthogonal contribution by introducing a truth-guided attention calibration mechanism.
>
> These updates ensure that our related work section accurately reflects the latest developments in hallucination mitigation research.
> We have included the analysis in Related Work and Appendix 15 of our revision.
>
> **Table 13: Hallucination Benchmark Comparison on COCO**
>
> *Hallucination rates (%) are reported using CHAIR$_S$ ($C_S$), CHAIR$_I$ ($C_I$), and BLEU (%) on COCO image captioning tasks, where lower CHAIR and higher BLEU are better. Intervention Target and Intervention Modality are reported with each method. The max new token is set to 512. The best results are highlighted in **bold**. $^\dagger$ represents results from original papers. $^\ddagger$ represents results from official code.*
>
> | Method | Intervention Target | Intervention Modality | $C_{S} \downarrow$ | $C_{I} \downarrow$ | Recall $\uparrow$ |
> |--------|-------------------|---------------------|-------------------|-------------------|------------------|
> | Baseline | - | - | 51.0 | 15.2 | 75.2 |
> | VTI [3] $^\dagger$ | Hidden State | Text & Image | 35.8 | 11.1 | 76.8 |
> | Nullu [4] $^\ddagger$ | Weights | Image | 54.6 | 14.1 | **81.3** |
> | HALC [5] $^\ddagger$ | Logits | Image | 41.8 | 12.2 | 80.3 |
> | **Ours** | **Attention** | **Text & Image** | **35.6** | **10.8** | 77.2 |
> | DeGF [8] $^\ddagger$ | Logits | Image | 46.0 | 14.2 | 78.5 |
> | **Ours + DeGF** | Attention & Logits | Text & Image | **45.8** | **14.0** | **79.3** |
> | AGLA [6] $^\dagger$ | Logits | Image | 43.0 | 14.1 | 78.9 |
> | **Ours + AGLA** | Attention & Logits | Text & Image | **40.1** | **13.0** | **81.2** |
> | Deco [7] $^\dagger$ | Hidden State | Image | 37.8 | 11.1 | **77.6** |
> | **Ours + Deco** | Attention & Hidden State | Text & Image | **32.6** | **10.9** | 77.5 |
>
> ## References
> [3].Reducing hallucinations in vision-language models via latent space steering ICLR 2024
>
> [4].Nullu: Mitigating Object Hallucinations in Large Vision-Language Models via HalluSpace Projection CVPR 2025
>
> [5].Halc: Object hallucination reduction via adaptive focal-contrast decoding ICML 2024
>
> [6].AGLA: Mitigating Object Hallucinations in Large Vision-Language Models with Assembly of Global and Local Attention CVPR 2024
>
> [7].MLLM can see? Dynamic Correction Decoding for Hallucination Mitigation ICLR 2024
>
> [8].Self-Correcting Decoding with Generative Feedback for Mitigating Hallucinations in Large Vision-Language Models ICLR 2025
>
> [9]. Object hallucination in image captioning EMNLP 2018

---

> > ### Comment · Reviewer_yxZC · 2025-11-25
> >
> > Thanks for your response and the additional results. However, after reading all the reviews and rebuttals, I agree that the main method still needs further revision to incorporate the improvements mentioned in the rebuttal, particularly regarding dynamic adaptation, robustness to layer selection, and applicability to other domains. The paper could be stronger with these improvements, so I will maintain my score.
> >
> > It is also unclear to me how to interpret the results in Table 5. The numbers seem to suggest that early-stage intervention works best, but I cannot tell from the table whether the generation degrades due to the intervention of low-level features.
> >
> > Also another minor concern, it is also unclear how the method would perform on more up-to-date LVLMs (e.g., Qwen2.5-VL).

---

> > > ### Author Response · Authors · 2025-11-25
> > >
> > > **Response to The numbers seem to suggest that early-stage intervention works best:**
> > >
> > > Thank you for the suggestion.
> > > To provide a clearer and more intuitive response to your observation that “early-stage intervention seems to work best,” we further compute the mean and variance of both  $C_S$ and BLEU for each model at each intervention stage. These statistics reflect, respectively, the average performance and stability of the model under each setting.
> > >
> > > As shown in the **Table 14** below, although MiniGPT-4 exhibits slightly lower $C_S$ in the early stage, its variance is extremely high across both $C_S$ and BLEU. This instability arises because early-stage intervention frequently triggers layer collapse (Layers 1–6, see Table 5), causing sharp fluctuations in performance. Thus, despite occasional strong results, early-stage intervention carries a substantial risk of destabilizing the model.
> > > A similar pattern is observed in LLaVA: although early-stage $C_S$ reach 14.15, the variance is again large, reaching 68.13 and 45.52, indicating unreliable behavior.
> > >
> > > In contrast, the middle stage exhibits considerably more stable performance across all three models.
> > > However, because middle layers are still relatively far from the final decoding layers, their intervention effect is weaker compared to the late stage under comparable BLEU conditions.
> > > For example, the mean $C_S$ for LLaVA is 19.42 (Table 6) in the middle stage but  17.28 in the late stage. Likewise, MPlug-Owl2 achieves 21.52 in the middle stage but 20.73 in the late stage.
> > >
> > > For these reasons, we select the late stage as the primary intervention point:
> > >
> > > (1) It achieves the best overall trade-off between hallucination mitigation and general performance;
> > >
> > > (2) It ensures greater stability and avoids the high-risk behavior seen in early-stage intervention.
> > >
> > >
> > >
> > > **Table 14: Comparison of Mean/Variance of CHAIR$_S$ and BLEU with Different Attention Intervention Stages**
> > >
> > > *The best results are in **bold**.*
> > >
> > > | Method | MiniGPT-4-7B | | | | LLaVA-1.5-7B | | | | mPLUG-Owl2-7B | | | |
> > > |--------|--------------|-|-|-|--------------|-|-|-|---------------|-|-|-|
> > > | | $\bar{C_{S}}\downarrow$ | $\text{Var}\_{C_{S}}\downarrow$ | $\bar{\text{BLEU}}\uparrow$ | $\text{Var}_{BLEU}\downarrow$ | $\bar{C_{S}}\downarrow$ | $\text{Var}\_{C_{S}}\downarrow$ | $\bar{\text{BLEU}}\uparrow$ | $\text{Var}_{BLEU}\downarrow$ | $\bar{C_{S}}\downarrow$ | $\text{Var}\_{C_{S}}\downarrow$ | $\bar{\text{BLEU}}\uparrow$ | $\text{Var}_{BLEU}\downarrow$ |
> > > | Early Stage | 8.95 | 40.97 | 9.75 | 34.08 | 14.15 | 68.13 | 12.81 | 45.52 | 21.63 | 1.65 | 16.28 | 0.01 |
> > > | Middle Stage | 16.76 | 1.21 | 16.02 | 0.39 | 19.42 | 2.82 | 16.54 | 0.06 | 21.52 | 0.35 | 16.27 | 0.01 |
> > > | Last Stage | 16.84 | 3.59 | 15.90 | 0.02 | 17.28 | 0.64 | 16.57 | 0.01 | 20.73 | 1.33 | 16.26 | 0.01 |
> > > | **Ours** | **15.62** | - | 15.79 | - | **16.43** | - | 16.48 | - | **19.40** | - | 16.21 | - |
> > >
> > > **Response to perform on Qwen2.5-VL:**
> > >
> > > Thank you for the suggestion. We will evaluate AGE on Qwen2.5-VL as well. Due to time constraints, we kindly ask for your patience while we complete and validate these additional experiments.

---

> ### Author Response · Authors · 2025-11-28
>
> Thank you for your suggestion. We additionally evaluated AGE on Qwen2.5-VL, using the Qwen2.5-VL-7B-Instruct-AWQ model with a maximum new-token length of 64, and tested it on the COCO Image Captioning benchmark. As shown in **Table 15** below, integrating AGE yields a 0.9% improvement in the CHAIR$_S$ metric, while maintaining the BLEU score.
>
> These results further verify that AGE is not a method that works only on specific datasets or particular model families. Instead, across LLaVA, MiniGPT-4, mPLUG-Owl2, and now Qwen2.5-VL, AGE consistently demonstrates a **general, reproducible, and model-agnostic attention calibration mechanism** that imitates the attention behavior of truthful tokens. This reflects the universal nature of the corrective signal encoded by AGE, rather than dataset-specific or architecture-specific tuning.
>
> Thank you again for your valuable feedback.
>
> **Table 15: Comparison of CHAIR and BLEU with Qwen2.5-VL as Baseline**
>
> *The best results are in **bold**.*
>
> | Method | $C_{S} \downarrow$ | $C_{I} \downarrow$ | BLEU $\uparrow$ |
> |--------|-------------------|-------------------|----------------|
> | Qwen2.5-VL | 13.0 | 5.9 | 12.3 |
> | **Qwen2.5-VL+Ours** | **12.1** | **5.3** | **12.5** |

---

### Official Review · Reviewer_Y1U5 · 2025-10-31

**Soundness:** 2
**Presentation:** 2
**Contribution:** 2
**Rating:** 2
**Confidence:** 4

**Summary:**

This paper presents AGE (Attention-aware Truth-Guided Enhancement), a training-free, decoding-time framework aimed at reducing hallucinations in large vision-language models (LVLMs). It leverages an in-depth analysis of attention patterns distinguishing truthful from hallucinated tokens across layers and modalities, applying targeted, stage-specific interventions to realign attention toward grounded responses. The approach is evaluated on benchmarks including COCO (with CHAIR metric), POPE, and MME, across three LVLMs (LLaVA, MiniGPT-4, mPLUG-Owl2), showing reduced hallucination rates while preserving or enhancing overall performance metrics like BLEU.

**Strengths:**

1. AGE offers a lightweight, training-free, and model-agnostic solution that can be integrated seamlessly during inference, making it practical for deployment.

2. The experimental setup is comprehensive, covering three diverse LVLMs and multiple benchmarks (COCO/CHAIR, POPE, MME), with results indicating hallucination mitigation alongside maintained or improved generation quality.

**Weaknesses:**

1. The core idea of using attention calibration or intervention to mitigate hallucinations by aligning with truthful patterns has been explored in several prior works, such as AGLA, which assembles global and local attention to reduce object hallucinations, DeCo, which employs dynamic correction decoding, and DeGF, which uses generative feedback for self-correction. Token-level and layer-wise analyses of hallucinations are also featured in these and related studies. While AGE contributes through finer-grained, stage-specific interventions, the paper does not sufficiently delineate its novelty beyond these approaches and lacks citations or direct comparisons to them.

2. The intervention stages (e.g., layer 20 for text, 30–31 for vision) are determined empirically (Appendix A.5), but the paper provides limited guidance for adapting these to novel architectures or LVLM variants. It notes that early-layer interventions severely degrade BLEU scores yet fails to propose a systematic methodology for identifying optimal stages or assessing transferability to deeper or wider models.

3. The formulation of δ as a "universal" intervention vector is empirically motivated and tuned, but lacks theoretical grounding on its effectiveness, particularly under distribution shifts between calibration and test data.

4. Notation in equations for attention weights exhibits inconsistencies, reported baseline metrics appear inconsistent with those in the original papers, potentially leading to unfair comparisons.

5. The method relies on numerous empirically set hyperparameters, with insufficient analysis of their robustness or generalizability across different settings.

**Questions:**

1. Given the similarities to prior works like AGLA, DeCo, and DeGF, could you clarify the novel contributions of AGE and provide comparisons (both qualitative and quantitative) to these methods? Additionally, please address discrepancies in baseline metrics compared to their original reports to ensure fair benchmarking.

2. Could you provide examples of failure cases for AGE, including their common characteristics (e.g., specific image types, prompt complexities, or distributional shifts)? This would help assess the method's limitations more concretely.

---

> ### Author Response · Authors · 2025-11-21
>
> **Response to W1 and Q1 about discussion with other methods (both qualitative and quantitative):** Thank you for raising this important point. Below, we clarify the conceptual distinctions between AGE and prior attention-based hallucination mitigation methods, and we provide direct empirical comparisons.
>
> **Discussion about related work:**
>
> **(1) AGLA:** Mitigates hallucinations by assembling global visual features for generation and local features for discrimination, addressing insufficient visual grounding. In contrast, AGE shows that hallucinations do not arise solely from weak visual attention—insufficient textual grounding also induces hallucinations. Moreover, AGE provides the first token-level, stage-specific visualization of how different layers emphasize image vs. text information, revealing model-intrinsic "truthful attention patterns" that prior work does not analyze. This yields clearer guidance on where and how to intervene.
>
> **(2) DeCo:** Focuses on comparing logits between earlier and later layers, assuming that earlier layers exhibit higher confidence for ground-truth tokens. Our findings differ in two key ways. Firstly, AGE uncovers that the root cause lies in mismatched image/text attention dynamics before token generation, rather than logit confidence. Secondly, while DeCo attributes hallucinations mainly to overloaded language priors suppressing visual signals, AGE shows that blindly adding visual attention is not always optimal—different models require different balances between text and image attention. This explains structural differences across LVLMs and motivates our stage-specific interventions.
>
> **(3) DeGF:** Augments inputs with externally synthesized visual information from generative models. AGE, however, argues that hallucination is fundamentally a layer-wise attention misalignment problem, not a missing-image-information problem. We identify which layers fail to follow real-token attention patterns and intervene precisely on those attention pathways, without auxiliary models or data.
>
> **Experiments:**
>
> 1. We directly compare AGE with AGLA, DeCo, and DeGF, and also integrate AGE into each system. As shown in **Table 13** below, AGE alone achieves the best or comparable performance, and when combined with AGLA/DeCo/DeGF, it consistently improves them in most of settings. This confirms that AGE introduces non-overlapping, complementary gains, supporting that our layer-wise, truth-guided attention imitation captures a fundamentally different source of hallucination.
>
> 2. Additionally, we originally did not compare AGE with HALC [5] because HALC relies on an external detector (Grounding DINO), which introduces additional supervision beyond our method. For completeness, we now include HALC in **Table 13**. The results show that AGE—despite requiring no auxiliary models—still achieves SOTA performance. This further demonstrates the effectiveness and independence of our hallucination-to-real-attention calibration.
>
> 3. For qualitative analysis, we include two visualization studies in **Fig. 14** and **Fig. 15**. As the results show, DeGF, DeCo, and AGLA all hallucinate the same nonexistent objects—such as the "chair" in **Fig.14** and the "wine glasses" and "cups" in **Fig. 15**. In contrast, AGE produces descriptions that are both more accurate and more detailed, offering a clearer and more intuitive demonstration of its effectiveness in reducing hallucinations.
>
> In summary, AGE differs from prior approaches by **(1)** identifying the token-level, stage-specific attention patterns that characterize truthful responses, **(2)** revealing model-dependent text-image attention requirements, and **(3)** intervening directly along these dynamics. These insights form the core conceptual novelty of our work.

---

> ### Author Response · Authors · 2025-11-21
>
> **Table 13: Hallucination Benchmark Comparison on COCO**
>
> *Hallucination rates (%) are reported using CHAIR$_S$ ($C_S$), CHAIR$_I$ ($C_I$), and BLEU (%) on COCO image captioning tasks, where lower CHAIR and higher BLEU are better. Intervention Target and Intervention Modality are reported with each method. The max new token is set to 512. The best results are highlighted in **bold**. $^\dagger$ represents results from original papers. $^\ddagger$ represents results from official code.*
>
> | Method | Intervention Target | Intervention Modality | $C_{S} \downarrow$ | $C_{I} \downarrow$ | Recall $\uparrow$ |
> |--------|-------------------|---------------------|-------------------|-------------------|------------------|
> | Baseline | - | - | 51.0 | 15.2 | 75.2 |
> | VTI [3] $^\dagger$ | Hidden State | Text & Image | 35.8 | 11.1 | 76.8 |
> | Nullu [4] $^\ddagger$ | Weights | Image | 54.6 | 14.1 | **81.3** |
> | HALC [5] $^\ddagger$ | Logits | Image | 41.8 | 12.2 | 80.3 |
> | **Ours** | **Attention** | **Text & Image** | **35.6** | **10.8** | 77.2 |
> | DeGF [8] $^\ddagger$ | Logits | Image | 46.0 | 14.2 | 78.5 |
> | **Ours + DeGF** | Attention & Logits | Text & Image | **45.8** | **14.0** | **79.3** |
> | AGLA [6] $^\dagger$ | Logits | Image | 43.0 | 14.1 | 78.9 |
> | **Ours + AGLA** | Attention & Logits | Text & Image | **40.1** | **13.0** | **81.2** |
> | Deco [7] $^\dagger$ | Hidden State | Image | 37.8 | 11.1 | **77.6** |
> | **Ours + Deco** | Attention & Hidden State | Text & Image | **32.6** | **10.9** | 77.5 |
>
> **Response to W2 about the guidance of layer chosen and identifying optimal stages:** Thank you for your insightful comments. Please see the **global response 2** for the layer selection and **global response 1** about identifying optimal stages dynamically.
>
> **Response to W2 about the Transferability to deeper/wider architectures:** Table 7 shows that AGE also improves LLaVA-13B (deeper and wider), decreasing $C_S$ by 3.2 and increasing BLEU. This confirms that the stage-wise attention patterns are effective across different architectures and that AGE transfers well across LVLM scales.
>
> **Response to W3 about effectiveness, particularly under distribution shifts:** Thank you for raising this important point. Please see the **global response 3** about the generalization AGE in domain shift benchmark.
>
> **Response to W4 about performance inconsistencies:** Thank you for highlighting the issue regarding baseline metrics—this is indeed crucial for ensuring fair and transparent evaluation. We clarify our sourcing strategy as follows.
>
> We report baseline metrics from two sources: [5] and the original papers:
> - Scores marked with $^\ddagger$ are taken directly from their respective original publications
> - Scores marked with $^\dagger$ come from the standardized and comprehensive benchmarking study of [5]
>
> We primarily adopt the standardized benchmark results ($^\dagger$) for two key reasons:
> 1. They ensure all models—including VCD, OPERA, and our baselines—are evaluated under identical implementation settings, decoding strategies, and maximum token lengths, which eliminates biases arising from inconsistent experimental setups
> 2. The standardized benchmark provides a complete set of metrics across all models, whereas original papers often omit important metrics (e.g., BLEU), making direct comparison incomplete or impossible
>
> This approach guarantees that the reported performance differences are fair, consistent, and verifiable.
>
> **Response to W4 about the notations:** Thanks for your valuable feedback. We fix the $I_{\text{hall}}^{(i)}$ in Eq. 3 to $I_{\text{real}}^{(i)}$.
>
> **Response to W5 about analysis of robustness or generalizability:**
> Thank you for pointing this out.
>
> - Please see the **global response 1** for the robustness and automatic selectivity for weight hyperparameters
> - Please see the **global response 2** for the robustness and generalizability for layer selection
> - Please see the **global response 3** for the robustness and generalizability of our setting generalized to domain shift benchmarks

---

> ### Author Response · Authors · 2025-11-21
>
> **Response to Q2 about failure cases:**
>
> Thank you for pointing this out.
>
> We further examine the failure cases—images that still exhibit hallucinations after applying AGE, and observe a consistent pattern: as shown in Fig. 13, AGE's calibration ability diminishes in scenes containing a large number of distinct small objects.
>
> In such complex settings, the abundance of visual instances increases the difficulty of accurate grounding, leading the model to continue producing hallucinations. We hypothesize that these cases require more fine-grained extraction and reinforcement of visual cues, which offers a clear direction for future improvements.
>
> ## References
> [3].Reducing hallucinations in vision-language models via latent space steering ICLR 2024
>
> [4].Nullu: Mitigating Object Hallucinations in Large Vision-Language Models via HalluSpace Projection CVPR 2025
>
> [5].Halc: Object hallucination reduction via adaptive focal-contrast decoding ICML 2024
>
> [6].AGLA: Mitigating Object Hallucinations in Large Vision-Language Models with Assembly of Global and Local Attention CVPR 2024
>
> [7].MLLM can see? Dynamic Correction Decoding for Hallucination Mitigation ICLR 2024
>
> [8].Self-Correcting Decoding with Generative Feedback for Mitigating Hallucinations in Large Vision-Language Models ICLR 2025
>
> [9]. Object hallucination in image captioning EMNLP 2018

---

> ### Author Response · Authors · 2025-11-25
> **Thanks again**
>
> Dear Reviewer Y1U5,
>
> Thank you again for taking the time to review our work and for the helpful comments you provided. We’ve carefully addressed all your concerns in our response and have updated the paper accordingly.
> We genuinely hope that our clarifications help reflect the real strength and intention of our work.
> We would sincerely appreciate it if you could consider updating your score to reflect the clarified contributions. If anything remains unclear or if you have further questions, we would be more than happy to continue the discussion at any time.
> Thank you again for your thoughtful review and engagement.
>
> Warm regards,
>
> The Authors

---

### Official Review · Reviewer_s179 · 2025-10-31

**Soundness:** 2
**Presentation:** 1
**Contribution:** 2
**Rating:** 2
**Confidence:** 4

**Summary:**

The goal of the paper is to address the hallucination in LVLMs. The paper conducts the layer-wise analysis by introducing the concept of Average Attention Sum and computing the difference between non-hallucinated and hallucinated ones. The analysis shows the patterns of $Diff^{l}_{image}$, which is positive at the last stage. The paper proposes an attention direction to imitate the attention pattern, which is used during inference. The paper validates the proposed method on hallucination benchmarks and conducts an ablation study, including sample size and variants of AGE.

**Strengths:**

1. The inference cost is lower than that of other methods, such as VCD. Once the attention direction is computed to imitate attention, inference requires only a simple shift operation.
2. Latent steering is widely adopted to mitigate hallucinations. But attention steering is a novel approach.

**Weaknesses:**

1. There is a logical gap in the hypothesis of the layer-wise analysis.
- Both MiniGPT-4 and mPLUG-Owl2 exhibit a consistently positive difference across all layers. Given this observation, is it valid to imitate Early and Late attentions for MiniGPT-4 and mPLUG-Owl2?
- In contrast, the difference in text is negative in general. This suggests that the corresponding $\lambda_t$ should be negative to reflect this. However, the paper assigns the positive value to $\lambda_t$.
2. The way to select the final hyperparameters, including the choice of layers and the scaling factor, is not clear. Details on the hyperparameter tuning method would be helpful.
3. The proposed approach steers the attention after computing the attention direction, which is similar to the latent steering methods [VTI, Nullu]. A detailed discussion and comparison are needed to validate the advantages of the proposed method.

**Questions:**

1. In Sec. 3.2, do humans manually annotate tokens as 'real/hallucinated', or is it done automatically?
2. In Eq. (3), the left term is divided by $I^{(i)}_{hall}$. Is it correct?
3. What common characteristics between MiniGPT and mPLUG-Owl lead to the positive $Diff^{l}_{image}$ for all layers?
4. After applying the proposed method to LVLMs, are the general task capabilities preserved?
5. After steering the attention, does the total attention still sum to 1?

---

> ### Author Response · Authors · 2025-11-21
>
> **Response to W1 about logical gap in the hypothesis of the layer-wise analysis:**
>
> We kindly remind you that our method follows a consistent rule: we only apply intervention ($\lambda > 0$) when the attention of real tokens is empirically higher than that of hallucinated tokens (i.e., a positive gap, Real $>$ Hallucination). If the gap is negative or negligible, we do not intervene ($\lambda = 0$). Below we will address your concerns from two views.
>
> **(1) Why not imitate Early Image Attention?**
>
> We observed that MiniGPT-4 and mPLUG-Owl2 exhibit a consistently positive attention gap (Real $>$ Hallucinated) across all layers, which suggests that imitating early attention might be another possible intervention strategy. To evaluate the strategy, as shown in our ablation study (Table 5), we compare the performance of ours with different intervention layers. We find that intervening in these shallow layers sometimes can lead to severe performance degradation (e.g., near-zero BLEU scores). The possible reason is that early layers (0-16) usually play as the role of fundamental visual feature extractors, and intervening these layers may hurt the ability for extracting the fundamental visual features. Therefore, despite the positive gap, we restrict the intervention to Late layers, where the model performs high-level semantic reasoning and object grounding, making the intervention both effective and safe.
>
> **(2) Is $\lambda_t$ positive despite the negative text gap?**
>
> The reviewer's observation that "text difference is generally negative" is correct for MiniGPT-4 and mPLUG-Owl2, but not correct for LLaVA-1.5. Specifically:
>
> - For **MiniGPT-4 and mPLUG-Owl2**, the text attention difference is generally negative (Real $<$ Hallucinated in text attention). Consistent with this, we do not apply text intervention ($\lambda_t=0$) for these models, aligning perfectly with the hypothesis.
> - For **LLaVA-1.5**, this model is unique. Due to its specific architecture (shallower visual fusion), its Middle Stage (Layers 16-26) relies heavily on linguistic context for reasoning. Our analysis (Section 3.3 and Figure 2 (a)) shows that in this specific stage, the text attention difference is generally positive (Real $>$ Hallucinated in text attention in middle stage).
>
> Thus, we only assign positive $\lambda_t$ for LLaVA to reinforce this stage-specific truth-grounding behavior in the middle stage. Therefore, there is no logical inconsistency between the intervention choices across models.
>
> **Response to W2 about hyperparameter:** Thank you for your suggestion. Please see the **global response 1** about weight hyperparameter selection, and **global response 2** for the layer selection.
>
> **Response to W3 about discussion and comparison to latent steering methods:**
>
> Thank you for your suggestion.
>
> - **VTI [3]:** Improves robustness by perturbing raw images with diverse noise patterns and stabilizing the latent visual representations. In contrast, our AGE directly imitates truthful attention patterns by computing a token-level directional vector between real and hallucinated responses, and uses it to calibrate both visual and textual attention during inference. Thus, VTI enhances the diversity of input features, whereas AGE adjusts attention directions in the decoding process—two fundamentally different mechanisms.
>
> - **Nullu [4]:** Analyzes distinctions between truthful and hallucinated responses, but its intervention operates at the sample level and manipulates the model's weight space. Our approach instead performs token-level analysis and identifies stable "real-token" attention signatures across model architectures and network depths, using these signatures to guide inference-time attention calibration. This distinction—sample-level weight adjustment versus token-level attention imitation—defines the core methodological difference.
>
> To further clarify non-overlap, we compare AGE with VTI and NullU using LLaVA as the backbone. We additionally evaluate against HALC [5], AGLA [6], Deco [7], and DeGF [8], and integrate AGE into these methods as a plug-in to demonstrate complementarity rather than redundancy.
>
> As shown in **Table 13** below, AGE achieves state-of-the-art performance among existing hallucination mitigation methods. Moreover, combining AGE with other approaches yields additional gains, confirming that our method offers a distinct and orthogonal contribution by introducing a truth-guided attention calibration mechanism. This demonstrates that AGE is not only non-duplicative but also provides a broadly generalizable improvement that can benefit diverse multimodal architectures.
> We have included the analysis in Appendix 15 of our revision.

---

> ### Author Response · Authors · 2025-11-21
>
> **Table 13: Hallucination Benchmark Comparison on COCO**
>
> *Hallucination rates (%) are reported using CHAIR$_S$ ($C_S$), CHAIR$_I$ ($C_I$), and BLEU (%) on COCO image captioning tasks, where lower CHAIR and higher BLEU are better. Intervention Target and Intervention Modality are reported with each method. The max new token is set to 512. The best results are highlighted in **bold**. $^\dagger$ represents results from original papers. $^\ddagger$ represents results from official code.*
>
> | Method | Intervention Target | Intervention Modality | $C_{S} \downarrow$ | $C_{I} \downarrow$ | Recall $\uparrow$ |
> |--------|-------------------|---------------------|-------------------|-------------------|------------------|
> | Baseline | - | - | 51.0 | 15.2 | 75.2 |
> | VTI [3] $^\dagger$ | Hidden State | Text & Image | 35.8 | 11.1 | 76.8 |
> | Nullu [4] $^\ddagger$ | Weights | Image | 54.6 | 14.1 | **81.3** |
> | HALC [5] $^\ddagger$ | Logits | Image | 41.8 | 12.2 | 80.3 |
> | **Ours** | **Attention** | **Text & Image** | **35.6** | **10.8** | 77.2 |
> | DeGF [8] $^\ddagger$ | Logits | Image | 46.0 | 14.2 | 78.5 |
> | **Ours + DeGF** | Attention & Logits | Text & Image | **45.8** | **14.0** | **79.3** |
> | AGLA [6] $^\dagger$ | Logits | Image | 43.0 | 14.1 | 78.9 |
> | **Ours + AGLA** | Attention & Logits | Text & Image | **40.1** | **13.0** | **81.2** |
> | Deco [7] $^\dagger$ | Hidden State | Image | 37.8 | 11.1 | **77.6** |
> | **Ours + Deco** | Attention & Hidden State | Text & Image | **32.6** | **10.9** | 77.5 |
>
> **Response to Q1 about the way of annotate tokens as 'real/hallucinated':**
>
> Thank you for the question.
>
> All token labels ("real" vs. "hallucinated") are obtained automatically, not manually. Following prior work [9], we use the COCO ground-truth object annotations as supervision. The procedure is:
>
> 1. **Extract noun tokens:** We first identify all noun tokens in the model's generated response using a POS-tagging parser. Only nouns are considered candidate visual entities.
> 2. **Map each noun to object categories:** Each noun is normalized—lowercased, lemmatized, and matched to COCO object categories using a synonym and alias dictionary (the same strategy as prior CHAIR-style evaluation).
> 3. **Compare against ground-truth annotations:** If a noun (or its synonym) appears in the COCO ground-truth object list for that image, it is labeled as a real token; otherwise, it is labeled as a hallucinated token.
> 4. **Fully automated pipeline:** No human annotators participate in token-level labeling.
>
> The entire process—noun extraction, synonym matching, and GT comparison—is fully automated and reproducible. A detailed description is provided in Appendix 10. This ensures that our analysis and construction of **${\delta}$** rely solely on standardized COCO annotations rather than manual human judgments.
>
> **Response to Q2 about left term in Eq.3:**
>
> Thanks for your valuable feedback. It should be $I_{real}^{(i)}$, we fix this in the paper and highlight with the color blue.
>
> **Response to Q3 about common characteristics between MiniGPT and mPLUG-Owl:**
>
> The difference mainly comes from the architectural and training discrepancies among LLaVA, MiniGPT-4, and mPLUG-Owl2.
>
> - **LLaVA** adopts a lightweight visual projector and a strongly text-driven instruction-tuning stage, which causes mid-layer representations to be dominated by linguistic reasoning, while visual grounding only re-emerges in late layers. Thus, LLaVA requires text-attention enhancement in middle layers and image-attention reinforcement in late layers.
>
> - **MiniGPT-4 and mPLUG-Owl2** employ deeper vision-language alignment modules (e.g., Q-former style or multi-layer projectors) and training pipelines that inject visual features more uniformly across all transformer layers. As a result, visual evidence consistently contributes throughout the entire reasoning stack, and hallucinations mainly stem from global under-attention to image tokens, not stage-specific imbalance.
>
> Therefore, common characteristics between MiniGPT and mPLUG-Owl lead to the positive image difference gap for all stages, while LLaVA exhibits stage-specific behavior due to its shallower fusion and more text-dominated mid layers.
>
> **Response to Q4 about general task ability after applying our method:**
>
> Yes, the model's general capabilities are well preserved.
>
> As shown in Table 2 and Fig. 4, our method maintains state-of-the-art performance on both POPE and MME, demonstrating strong discriminative ability. Additionally, to evaluate the generalization capability of AGE across unseen domains, we conduct extensive domain-shift experiments across multiple LVLMs. Please see the **global response 3** about generalization AGE in domain shift benchmark.

---

> ### Author Response · Authors · 2025-11-21
>
> **Response to Q5 about attention sum:** The total attention does not remain normalized to 1 after our intervention.
>
> In our analysis, we extract the model's post-softmax attention during inference to study the differences between real and hallucinated tokens. Accordingly, the calibration vector **${\delta}$** is also applied after softmax, directly adjusting the final attention distribution to steer it toward the real-token pattern. Because this modification occurs after normalization, the adjusted attention no longer sums to 1.
>
> Importantly, extensive evaluations across benchmarks (Table 2, Fig. 4 and Table 9) show that this does not hinder model inference; instead, it consistently improves performance. Our intervention modifies relative attention strengths while preserving the model's ability to aggregate context effectively, leading to more reliable generation.
>
> ## References
> [3].Reducing hallucinations in vision-language models via latent space steering ICLR 2024
>
> [4].Nullu: Mitigating Object Hallucinations in Large Vision-Language Models via HalluSpace Projection CVPR 2025
>
> [5].Halc: Object hallucination reduction via adaptive focal-contrast decoding ICML 2024
>
> [6].AGLA: Mitigating Object Hallucinations in Large Vision-Language Models with Assembly of Global and Local Attention CVPR 2024
>
> [7].MLLM can see? Dynamic Correction Decoding for Hallucination Mitigation ICLR 2024
>
> [8].Self-Correcting Decoding with Generative Feedback for Mitigating Hallucinations in Large Vision-Language Models ICLR 2025
>
> [9]. Object hallucination in image captioning EMNLP 2018

---

> ### Author Response · Authors · 2025-11-25
> **Thanks again**
>
> Dear Reviewer s179,
>
> Thank you again for taking the time to review our work and for the helpful comments you provided. We’ve carefully addressed all your concerns in our response and have updated the paper accordingly.
> We genuinely hope that our clarifications help reflect the real strength and intention of our work.
> We would sincerely appreciate it if you could consider updating your score to reflect the clarified contributions. If anything remains unclear or if you have further questions, we would be more than happy to continue the discussion at any time.
> Thank you again for your thoughtful review and engagement.
>
> Warm regards,
>
> The Authors

---

### Official Review · Reviewer_sf9o · 2025-10-31

**Soundness:** 4
**Presentation:** 4
**Contribution:** 4
**Rating:** 8
**Confidence:** 3

**Summary:**

The paper proposes AGE, a training-free, decoding-time framework that mitigates hallucinations in LVLMs by imitating the stage-specific attention patterns observed in truthful tokens. From a token- and layer-wise analysis, the authors find that real tokens systematically allocate more visual attention in late layers, while some models (e.g., LLaVA-1.5) also require stronger text attention mid-stage. AGE injects a directional visual correction vector δ (estimated from a small set of COCO samples) into late-stage layers and applies self-multiplicative text attention in the mid-stage when model-specific analysis indicates it helps. Across COCO/CHAIR, POPE, and MME, AGE outperforms recent SOTA methods while maintaining or slightly improving BLEU, and requires no additional training. Ablations show the vector-guided visual intervention (AGEI) is substantially more effective than coarse self-scaling, and that targeted, stage-aware interventions beat blanket modifications. The method is simple, interpretable at the attention level, and appears robust to the number of samples used to estimate δ.

**Strengths:**

- A precise, stage-aware recipe—directional visual correction in late layers + optional mid-stage text boost—plugs into existing LVLMs without retraining or extra modules.
- The token-/layer-wise diagnosis directly motivates the interventions; ablations (SMA vs. AGEI, AGEM/AGEL variants) convincingly show why direction + stage targeting matter.
- Strong gains on CHAIR/POPE/MME across LLaVA-1.5, MiniGPT-4, mPLUG-Owl2, with BLEU preserved; δ estimated from M=10 samples yet still competitive—good efficiency story.

**Weaknesses:**

- The approach assumes attention patterns of “real” tokens are the right supervisory signal; while effective, this remains heuristic and may not universally reflect grounding quality across tasks or architectures.
- Fixed layer choices (20, 30–31) and large λᵥ=100 / λₜ=3 suggest potential sensitivity; it’s unclear how easily settings transfer to unseen models, domains (e.g., medical, diagram, chart), or alternative decoders.
 -“Real vs. hallucinated” token labeling depends on object annotations (COCO); generalization to open-ended VQA without dense GT, long-context reasoning, or video remains untested, and the latency/throughput overhead of per-step intervention isn’t reported.

**Questions:**

- How sensitive are results to the layer indices and λᵥ/λₜ? Could you automatically pick stages (e.g., by monitoring live attention gaps) to avoid manual choices?
- δ is computed from COCO captions—how does AGE transfer to domain-shifted settings (e.g., medical images, charts) or text-heavy VQA where “real vs. hall” labels are less object-centric?
- Have you considered online estimation of δ during decoding (or per-image calibration) to reduce dependence on an offline sample pool and improve robustness to image/content diversity?

---

> ### Author Response · Authors · 2025-11-21
>
> **Response to W1 about heuristic:** We appreciate this insightful comment. While identifying real-token attention patterns starts as an empirical observation, our extensive analysis confirms that this signal serves as a robust behavioral signature of truthful generation rather than a mere heuristic. We validate this utilizing two key findings:
>
> **(1) Universal Stability Across Architectures:** As analyzed in Section 3, diverse architectures (LLaVA, MiniGPT-4, mPLUG-Owl2) consistently exhibit distinct, stage-specific attention disparities between real and hallucinated tokens. This suggests the "attention gap" captures a fundamental behavioral divergence in multimodal reasoning rather than model-specific noise.
>
> **(2) Strong Transferability (Crucial Evidence):** To directly address the concern about universal validity, we applied the intervention vector $\delta$—derived solely from COCO to 5 out-of-domain benchmarks: Medical, Video, Math, Map, and Table.
>
> For details, please see Appendix 11 of our revised manuscript or our response to Weakness 2. The results indicate that our method has desired generalization ability to domain-shifted settings, effectively improves grounding quality across diverse tasks. It proves that the "real-token attention pattern" encodes a universal "corrective tendency" (re-balancing visual vs. textual reliance), verifying its validity as a supervisory signal.
>
> **Response to W2 and Q2 about domain-shifted settings**
> Thank you for the thoughtful feedback. Please refer to **global response 3** about generalization AGE in domain shift benchmark.
>
> **Response to W2 about latency/throughput**
> Regarding the concern about overhead, **Table 10** below shows that AGE introduces negligible additional latency and preserves high throughput across all tested models. We have included the analysis in Appendix 12 of our revision.
>
> **Table 10: Comparison of latency (seconds/tokens) and throughput (tokens/seconds) with different LVLMs**
>
> | Method | MiniGPT-4-7B | | LLaVA-1.5-7B | | mPLUG-Owl2-7B | |
> |--------|--------------|-|--------------|-|---------------|-|
> | | latency ↓ | throughput ↑ | latency ↓ | throughput ↑ | latency ↓ | throughput ↑ |
> | | (s/t) | (t/s) | (s/t) | (t/s) | (s/t) | (t/s) |
> | Baseline | 7.1e-5 | 14104.4 | **1.7e-4** | **5700.8** | 2.1e-4 | 4635.4 |
> | **Ours** | **6.8e-5** | **14646.9** | 1.8e-4 | 5418.6 | **1.9e-4** | **5234.2** |
>
> **Response to Q1 about layer indices, lambda and stage:** Thanks for your feedback. Please refer to our **global response 1** about sensitivity to $\lambda_v$ and $\lambda_t$ and automatically pick stages. Please refer to our **global response 2** about sensitivity to layer index.
>
> **Response to Q3 about online estimation $\delta$:** Thank you for the insightful feedback.
>
> We agree with you that fully online estimation is promising. However, without ground-truth supervision during inference, online estimation is challenging. Since the model cannot reliably distinguish real tokens from hallucinatory ones, making it impossible to compute an accurate **$\{\delta}$** vector as we do offline.
>
> Thus we devise a more direct online calibration mechanism that avoids using **$\{\delta}$**. Instead of relying on real-hallucination discrepancies, the online variant applies self-multiplicative enhancement to both image and text attention, allowing the model to internally amplify reliable signals.
>
> As shown in **Table 12** below, this online version does not match the performance of the offline **$\{\delta}$**-guided method, but it still achieves consistent and notable gains over the baseline, demonstrating that effective hallucination mitigation is possible even without offline supervision. These results further support the core motivation of our work: The model benefits from imitating truth-aligned attention behaviors, and this principle remains effective even when the calibration is performed online without explicit **$\{\delta}$**. We have included the analysis in Appendix 14 of our revision.
>
> **Table 12: Comparison between the AGE-Online variants on LLaVA-1.5**
>
> *The best results are shown in **bold**.*
>
> | Method | $C_{S} \downarrow$ | $C_{I} \downarrow$ | BLEU $\uparrow$ |
> |--------|-------------------|-------------------|----------------|
> | Baseline | 20.80 | 6.77 | 15.93 |
> | $AGE_{\text{Online}}$ | 16.93 | 6.05 | 16.13 |
> | **AGE** | **16.43** | **5.58** | **16.48** |

---

> > ### Comment · Reviewer_sf9o · 2025-11-21
> >
> > Thank you for the detailed response. However, after reviewing the weaknesses raised by the other reviewers and considering them collectively, I have decided to adjust my score.

---

> > > ### Author Response · Authors · 2025-11-21
> > >
> > > Thank you very much for your feedback. We kindly encourage you to reconsider our clarifications regarding the concerns raised by other reviewers, as several issues stem from misunderstandings. Please let us know the specific concerns or questions you have, so we can address them. We believe that our detailed responses will help you understand our contributions more clearly and retain your favorable consideration of our work.

---

> > > > ### Comment · Reviewer_sf9o · 2025-11-27
> > > >
> > > > Sorry for the delayed response. After reviewing the authors’ replies to the other reviewers, I realized that I had misunderstood part of the discussion. The detailed response has clarified the issue for me. Therefore, I will adjust my score back to the original rating. Thank you.

---

> > > > > ### Author Response · Authors · 2025-11-27
> > > > > **Appreciation for the Reviewer’s Recognition**
> > > > >
> > > > > Thank you very much for your recognition and for the time and valuable suggestions you have contributed during the review process.

---

### Author Response · Authors · 2025-11-21
**Global Response**

**Summary of Revisions & Additions:**

- **Hyperparameter Rationale & Automation**: Expended **Appendix 13** to demonstrate the rationale and introduced Dynamic AGE analyze.
- **Layer Selection Mechanism & Robustness**: Expanded ablations **Tables 5 & 6** for explanation to stage-based (not index-based) layer selection.
- **Domain-Shift Analyze**: Expended **Appendix 11, Table 9** to demonstrate that the learned “truth-aligned attention pattern’’ is transferable, not dataset-specific.
- **Online δ-Free Variant**: Added an additional online AGE method **Appendix 14, Table 12** that avoids using **δ**, demonstrates AGE’s central idea enables online hallucination mitigation.
- **Extended Comparisons**: Added detailed quantitative and qualitative comparisons vs. AGLA, DeCo, DeGF, VTI, Nullu in **Appendix 15, Table 13, Figures 14, 15** demonstrate the core conceptual novelty of our work.
- **Failure Case Analysis**: Expended **Appendix 16, Figure 13** for failure case study provides insight for future work on finer-grained grounding.

We thank all reviewers for their time and valuable feedback and suggestions, which are useful for improving our work. We first address the overarching concerns before providing point-by-point responses to each reviewer. Our detailed replies are presented in the following sections. All corresponding revisions in the manuscript have been highlighted in blue. We hope that these revisions adequately address the reviewers' comments and enhance the clarity and quality of our paper.

---

> ### Author Response · Authors · 2025-11-21
>
> ## Global Response 1: On the Rationale, Robustness, and Automation of Hyperparameters ($\lambda_v, \lambda_t$)
> *(Addresses concerns from Reviewers SF9O, S179, Y1U5, YXZC regarding sensitivity and manual tuning)*
>
> We clarify that the hyperparameters in AGE are statistically derived from model-intrinsic properties, validated for robustness, and capable of full automation. They are not arbitrary values found via random tuning.
>
> ### (1) Statistical Derivation (Physical Grounding of $\lambda$)
>
> The scaling factor $\lambda$ is calculated to align the magnitude of the intervention vector **$\{\delta}$** with the actual scale of the attention discrepancy (gap) observed between real and hallucinated tokens.
>
> **(a) Visual Attention ($\lambda_v$):**
> The vector **${\delta}$** is computed from the $M$ samples in our auxiliary set. To determine its effective strength, we verify the Root Mean Square (RMS) of **${\delta}$** against the layer-wise attention gap ($\text{Diff}^{l}$) observed in our analysis:
>
> $$\lambda_v = \frac{\text{Diff}^{l}}{RMS({\delta})}, \quad \text{where } RMS({\delta}) = \sqrt{\frac{1}{n}\sum_{i=1}^{n} \{\delta}_i^2}$$
>
> Taking LLaVA as an example, the observed attention gap $\text{Diff}$ in layers 30-31 is $\sim 3.3e^{-3}$, while the RMS strength of **${\delta}$** is $\sim 3.3e^{-5}$. This naturally yields a ratio of $\approx 100$, providing the theoretical basis for our default setting $\lambda_v=100$.
>
> **(b) Textual Attention ($\lambda_t$):**
> Similarly, $\lambda_t$ is derived from the ratio of real-to-hallucinated textual attention sums:
> $\lambda_t = \bar{s}_{(\text{real})} / \bar{s}\_{(\text{hall})} \approx 1.2$.
> In practice, we use a slightly stronger factor ($\lambda_t=3$) to enhance robustness, though performance is stable across values.
>
> ### (2) High Robustness (Insensitivity to Exact Values)
>
> Our method relies on the corrective direction of the intervention rather than precise scalar tuning. As shown in Figure 7, extensive sensitivity analyses demonstrate that AGE maintains consistent performance improvements over the baseline across a wide range of values ($\lambda_v \in [80, 120]$ and $\lambda_t \in [1, 5]$). This confirms that the method is stable and does not require fragile fine-tuning.

---

> ### Author Response · Authors · 2025-11-21
>
> ### (3) Fully Automated Variant (Dynamic AGE)
>
> To completely eliminate dependencies on manual layer selection and hyperparameter tuning, we further develop Dynamic AGE (see Appendix 13). This variant replaces fixed settings with an adaptive mechanism that monitors the live attention gap at every decoding step. The procedure consists of three phases:
>
> **Phase 1: Constructing Reference Profiles (Offline)**
>
> We first aggregate attention statistics from the $M$ auxiliary samples to build a "Hallucination Reference Profile," representing the typical attention state of hallucinated tokens:
>
> - **Visual Reference Vector ($a^*_{(hall)}$):** The weighted average of visual attention vectors from hallucinated tokens:
>   $$a_{(hall,vision)}^{*} = M \sum_{i=1}^{M} w_{i} \cdot (\overline{a}_{(hall,vision)}^{i})$$
>
> - **Textual Reference Sum ($s^*_{(hall)}$):** The weighted average of attention sums allocated to text tokens in hallucinated responses:
>   $$s_{(hall,text)}^{*} = M \sum_{i=1}^{M} w_{i} \cdot \overline{s}_{(hall,text)}^{i}$$
>
> **Phase 2: Calculating Dynamic Gaps (Online)**
>
> During inference, for each layer $l$ and decoding step $k$, we calculate how much the current attention state deviates from the hallucination profile:
>
> - **Visual Instantaneous Deviation ($\delta_d$):** We define $\delta_d$ as the vector difference between the current visual attention $a_{vision}^{l,k}$ and the hallucination reference. This captures the *live* drift of the current token:
>   $$\delta_{d}^{(l,k)} = a_{vision}^{l,k} - a_{(hall,vision)}^{*}$$
>
> - **Dynamic Visual Scaling ($g_v$):** We determine the intervention strength by normalizing the magnitude (RMS) of this instantaneous deviation ($\delta_d$) against the global calibration vector ($\delta=\frac{1}{M}\sum_{i=1}^{M}w_{i}\cdot(\mathbf{\overline{a}}\_{\text{(real,vision)}}^{i}-\mathbf{\overline{a}}\_{\text{(hall,vision)}}^{i})$, derived in Eq. 6). This ratio quantifies how "strong" the current deviation is relative to the expected correction direction:
>   $$g_{v}^{l} = \frac{RMS(\delta_{d}^{(l,k)})}{RMS(\delta)}$$
>
> - **Dynamic Textual Scaling ($g_t$):** Similarly, we measure the ratio of the current text attention sum to the hallucination reference sum:
>   $$g_{t}^{l} = \frac{s_{text}^{(l,k)}}{s_{(hall,text)}^{*}}$$
>
> **Phase 3: Adaptive Calibration**
>
> Crucially, intervention is triggered only when the dynamic gap implies a significant mismatch with truthful patterns (i.e., exceeding a statistical threshold $\tau$). The model automatically self-calibrates using the computed dynamic strengths:
>
> - **Visual Calibration:** If the dynamic gap $g_v^l$ exceeds $\tau_v$, we inject the global correction vector $\delta$ scaled by the live strength $g_v^l$:$$\hat{a}\_{vision}^{l} = a_{vision}^{l} + g_{v}^{l} \times \delta, \quad \text{if } g_{v}^{l} > \tau_v$$
>
> - **Textual Calibration:** If the textual attention ratio $g_t^l$ exceeds $\tau_t$ (indicating a need for stronger context grounding), we perform self-multiplicative enhancement:
>   $$\hat{a}\_{text}^{l} = a_{text}^{l} + g_{t}^{l} \times a_{text}^{l}, \quad \text{if } g_{t}^{l} > \tau_t$$
>
> **Result:** This mechanism enables the model to automatically determine both which layers to intervene in and how much modulation strength to apply at each step (via $g_v^l, g_t^l$), without any manual tuning.
>
> Following the logic used for $\lambda_v$ and $\lambda_t$, we initialize their base values to 100 and 1, respectively. The terms $\tau_v$ and $\tau_t$ capture the instantaneous discrepancy between the current token's attention and the hallucination pattern, and simultaneously act as dynamic weights that guide the calibration strength.
>
> We evaluate this dynamic variant of AGE in Table 11. While $AGE_{\text{Dynamic}}$ offers strong flexibility in both layer choice and weighting, its slightly higher $C_S$ and $C_I$ scores—relative to manually tuned AGE—suggest a modest performance trade-off. We attribute this to the more aggressive and unconstrained adjustments occasionally interfering with the preservation of low-level visual or textual features, reducing the net benefit of semantic-level calibration.
>
> Nevertheless, $AGE_{\text{Dynamic}}$ remains substantially better than the baseline across all metrics, demonstrating that the core mechanism of AGE is inherently robust, adaptive, and transferable, even without manually specified parameters.
>
> We have included these results into Appendix 13 of our revision.
>
> **Table 11: Comparison between the AGE-Dynamic variants on LLaVA-1.5**
>
> *The best results are shown in **bold**.*
>
> | Method | $C_{S} \downarrow$ | $C_{I} \downarrow$ | BLEU $\uparrow$ |
> |--------|-------------------|-------------------|----------------|
> | Baseline | 20.80 | 6.77 | 15.93 |
> | $AGE_{\text{Dynamic}}$ | 17.77 | 5.89 | **16.80** |
> | **AGE** | **16.43** | **5.58** | 16.48 |

---

> ### Author Response · Authors · 2025-11-21
>
> ## Global Response 2: On Layer Selection Strategy (Mechanism & Robustness)
> *(Addresses concerns from Reviewers SF9O, S179, Y1U5, YXZC regarding specific layer indices and why early layers are excluded)*
>
> Our layer selection is not based on fragile, hand-picked choices but follows a principled, stage-aware and empirically validated procedure.
>
> ### Layer selection is stage-based, not index-based
>
> To test robustness, we performed layer-by-layer intervention across all Transformer blocks:
>
> - **Image-attention intervention (Table 5):** evaluated on three LVLMs (MiniGPT-4, LLaVA, mPLUG-Owl2)
> - **Text-attention intervention (Table 6):** only LLaVA uses it, so results are reported accordingly
>
> As shown in Tables 5 and 6, any layer within our recommended stages (Middle stage for text attention, Late stage for vision attention) consistently improves performance, demonstrating that AGE is insensitive to the exact layer index.
>
> ### Why early-layer intervention is not recommended?
>
> Although some early layers occasionally show small improvements, we also observe frequent model collapse: *e.g.* MiniGPT-4 crashes at layers 3-6, LLaVA collapses at layers 1, 2, 5, 6.
>
> This matches prior analyses [3,4]:
> - Early layers encode low-level visual features
> - Modifying attention there disrupts fundamental feature extraction
> - This harms BLEU/CHAIR far more than it helps hallucination mitigation
>
> Thus, only semantic reasoning stages (middle/late) provide a safe intervention zone.
>
> We therefore select layers 20 (text) for LLaVA and 30–31 (vision) for MiniGPT-4, LLaVA, and mPLUG-Owl2, simply because they yield slightly more stable gains—not because AGE depends on these specific indices.
>
> **AGE is robust: what matters is the stage, not the exact layer.**
>
> Across all three models: late-stage vision intervention or middle-stage text intervention consistently improve hallucination metrics. AGE therefore shows robustness to layer selection, as long as the intervention is applied within the correct functional stage.
>
> Besides, we emphasize that selecting specific reasoning layers is a commonly-used paradigm in decoding-time intervention. For example, DoLA (which contrasts "premature" vs. "mature" layers) and DeCo (which relies on confidence gaps between early and late layers) also depend on identifying specific layer ranges where semantic information is most discriminative.
>
> Thus AGE does not rely on brittle layer choices. The method is governed by stable functional stages (semantic layers), not specific indices. Early-layer interventions are avoided because they damage fundamental visual/textual representations. Within the correct stage, AGE is robust to the exact layer choice.
>
> **Table 5: Comparison of CHAIR and BLEU with Different Image Attention Intervention Layers**
>
> *For LLaVA, we do not intervene with image attention in the Middle Stage; only intervene with text attention is needed, while other models intervene in all Stages. The best results are in **bold**.*
>
> | Method | MiniGPT-4-7B | | | LLaVA-1.5-7B | | | mPLUG-Owl2-7B | | |
> |-------|--------------|-|-|--------------|-|-|---------------|-|-|
> | | $C_{S} \downarrow$ | $C_{I} \downarrow$ | BLEU $\uparrow$ | $C_{S} \downarrow$ | $C_{I} \downarrow$ | BLEU $\uparrow$ | $C_{S} \downarrow$ | $C_{I} \downarrow$ | BLEU $\uparrow$ |
> |------|--------------|--------------|---------------|--------------|--------------|---------------|---------------|---------------|---------------|
> | baseline | 30.87 | 12.33 | 14.33 | 20.80 | 6.77 | 15.93 | 23.20 | 8.33 | 15.37 |
> | **Early Stage** | | | | | | | | | |
> | 1,2 | **8.80** | **4.73** | 12.38 | 0.00 | 0.00 | 1.03 | 22.20 | 8.55 | 16.12 |
> | 3,4 | 2.20 | 3.31 | 4.93 | 19.00 | 6.05 | 16.60 | 20.60 | 7.84 | 16.29 |
> | 5,6 | 2.00 | 15.85 | 3.68 | 0.20 | 1.03 | 2.96 | 20.40 | 8.15 | 16.25 |
> | 7,8 | 14.60 | 9.71 | 14.82 | 20.00 | 6.63 | **16.70** | 19.20 | 7.70 | 16.30 |
> | 9,10 | 13.80 | 9.44 | 14.60 | 19.00 | 6.36 | 16.50 | 21.80 | 8.20 | **16.34** |
> | 11,12 | 0.20 | 2.12 | 0.56 | 17.80 | 5.87 | 16.17 | 23.20 | 8.21 | 16.23 |
> | 13,14 | 18.80 | 7.61 | 15.64 | 17.80 | 5.96 | 16.43 | 22.60 | 8.66 | 16.38 |
> | 15,16 | 11.20 | 4.00 | 15.37 | 19.40 | 6.66 | 16.52 | 23.00 | 8.50 | 16.29 |
> | **Middle Stage** | | | | | | | | | |
> | 17,18 | 15.60 | 5.57 | **16.99** | - | - | - | 22.20 | 8.06 | 16.24 |
> | 19,20 | 15.80 | 6.12 | 15.78 | - | - | - | 20.80 | 7.56 | 16.30 |
> | 21,22 | 18.20 | 7.52 | 15.54 | - | - | - | 21.40 | 8.02 | 16.26 |
> | 23,24 | 17.20 | 5.78 | 16.12 | - | - | - | 21.20 | 7.97 | 16.29 |
> | 25,26 | 17.00 | 6.13 | 15.68 | - | - | - | 22.00 | 8.18 | 16.26 |
> | **Late Stage** | | | | | | | | | |
> | 27,28 | 15.91 | 6.19 | 15.86 | 18.00 | 6.10 | 16.60 | 21.40 | 8.03 | 16.26 |
> | 29,30 | 19.00 | 6.76 | 16.05 | 17.40 | 5.75 | 16.64 | 21.40 | 7.99 | 16.32 |
> | **Ours (30,31)** | 15.62 | 6.00 | 15.79 | **16.43** | **5.58** | 16.48 | **19.40** | **7.47** | 16.21 |

---

> ### Author Response · Authors · 2025-11-21
>
> ## Global Response 3: The generalization of AGE in domain shift benchmark
> *(Addresses concerns from Reviewers SF9O, S179, Y1U5, YXZC regarding **${\delta}$** as a 'universal' vectors transfer to unseen domains)*
>
> We conduct extensive domain-shift experiments across multiple LVLMs. Specifically, we directly apply the $\delta$ computed on COCO to five out-of-domain settings, including Medical [1], Video, Math, Map, and Table [2], which differ substantially from natural images.
>
> - **Medical benchmark:** Follow [1] and report Accuracy under FAKE and SWAP settings
> - **Other domains (Video, Math, Map, Table):** Report Accuracy/F1 following [2]
>
> As shown in **Table 9** bellow, AGE consistently improves over the baseline across all domains and architectures (*e.g.*, +3.2% in Medical for LLaVA, +3.8% in Math for MiniGPT-4, +1.6% in Table for mPLUG-Owl2).
>
> These results demonstrate that the calibrated hallucination-to-truth signal encoded by $\delta$ and the chosen stages remain effective even under severe distribution shift, indicating strong robustness and transferability rather than dataset-specific tuning.
>
> Our domain-shift experiments—covering medical images, math diagrams, tables, and video frames—confirm that AGE generalizes well even in tasks without object-level supervision. We have included these results into Appendix 11 of our revision.
>
>
> **Table 6: Comparison of CHAIR and BLEU with Different Text Attention Intervention Layers in LLaVA**
>
> *The best results are in **bold**.*
>
> | Layer | $C_{S} \downarrow$ | $C_{I} \downarrow$ | BLEU $\uparrow$ |
> |-------|-------------------|-------------------|----------------|
> | Baseline | 20.8 | 6.7 | 15.9 |
> | 16 | 20.6 | 7.0 | 16.0 |
> | 17 | 22.6 | 7.5 | 16.8 |
> | 18 | 19.0 | 6.3 | 16.5 |
> | 19 | 20.0 | 6.5 | 16.2 |
> | **20** | **16.4** | **5.5** | 16.4 |
> | 21 | 20.0 | 7.1 | 16.7 |
> | 22 | 18.8 | 6.0 | **16.8** |
> | 23 | 17.0 | 5.7 | 16.5 |
> | 24 | 20.0 | 6.4 | 16.5 |
> | 25 | 19.8 | 7.1 | 16.7 |
> | 26 | 19.4 | 6.5 | 16.8 |
>
>
>
> **Table 9: Cross-domain Benchmark Results**
>
> *Note: Source **$\{\delta}$** is generated by COCO. Comparison with different domain benchmarks: Medical [1], Video [2], Math [2], Map [2], and Table [2]. FAKE question: Fake or nonsensical questions are used to examine model's ability to detect incoherent questions. Image SWAP: In this scenario, we swap the images with unrelated ones to evaluate the model's ability to detect mismatches between the image content and the question. The best results are highlighted in **bold**.*
>
> | Methods | Medical-FAKE ↑ | Medical-NONE ↑ | Video-Acc ↑ | Video-F1 ↑ |
> |---------|---------------|---------------|------------|-----------|
> | LLaVA | 20.9 | 5.8 | 57.7 | 53.2 |
> | **Ours** | **24.5** | **8.6** | **58.7** | **54.0** |
>
> | Methods | Math-Acc ↑ | Math-F1 ↑ | Map-Acc ↑ | Map-F1 ↑ |
> |---------|-----------|----------|----------|----------|
> | MiniGPT-4 | 66.6 | 40.0 | 25.0 | 24.9 |
> | **Ours** | **68.2** | **46.0** | **26.9** | **26.8** |
>
> | Methods | Table-Acc ↑ | Table-F1 ↑ | Map-Acc ↑ | Map-F1 ↑ |
> |---------|------------|-----------|----------|----------|
> | mPLUG-Owl2 | 72.7 | 52.2 | 36.5 | 41.4 |
> | **Ours** | **74.5** | **53.5** | **41.3** | **44.8** |
>
> ## References
> [1].Hallucination Benchmark in Medical Visual Question Answering  ICLR 2024
>
> [2].HallusionBench: An Advanced Diagnostic Suite for Entangled Language Hallucination and Visual Illusion in Large Vision-Language Models Proceedings of the IEEE/CVF Conference on Computer Vision and Pattern Recognition (CVPR) 2024
>
> [3]. Instruction-Guided Fusion of Multi-Layer Visual Features in Large Vision-Language Models Pattern Recognition 2024
>
> [4]. Multimodal Language Models See Better When They Look Shallower EMNLP 2025

---

### Meta-Review · Area_Chair_6XfV · 2025-12-18

**Summary:**

The paper presents a novel method called AGE (Attention-aware Truth-Guided Enhancement) to mitigate hallucinations in large vision-language models (LVLMs) without requiring retraining. AGE leverages a detailed, stage-specific analysis of attention patterns in the model to calibrate the attention towards grounded responses. The approach is evaluated across multiple LVLMs (LLaVA, MiniGPT-4, mPLUG-Owl2) and benchmarks (COCO, POPE, MME), showing significant improvements in hallucination mitigation while preserving or enhancing other performance metrics like BLEU. The key advantage of AGE lies in its simplicity, efficiency, and model-agnostic nature, making it a compelling solution for practical deployment in various LVLMs. Overall, the authors have adequately addressed the reviewers’ concerns and provided substantial clarifications and additional results. Thus, I recommend acceptance of this submission.

**Reviewer Concerns:**

The authors addressed the concerns raised by all reviewers in a detailed and comprehensive manner. Specifically:

• Reviewer s179 had concerns about the layer-wise analysis and intervention strategy, particularly regarding the positive attention gap in early layers. The authors clarified that their method only intervenes when a positive gap is observed between real and hallucinated tokens, and this intervention is applied selectively to later layers for stability. This explanation directly addressed the reviewer’s concerns.

• Reviewer Y1U5 questioned the novelty of AGE compared to prior works (AGLA, DeCo, DeGF) and raised concerns about hyperparameter selection. The authors provided a clear, point-by-point distinction between their approach and these prior methods, emphasizing the unique contributions of AGE, such as token-level and stage-specific intervention, which were not addressed by previous work. The authors also clarified the robustness of their hyperparameter choices, addressing the reviewer's concerns about generalizability.

• Reviewer sf9o raised concerns about the method’s reliance on pre-calculated attention vectors and its sensitivity to layer indices. The authors further clarified that their approach is not dependent on early-layer interventions due to stability issues and that AGE has been evaluated across a variety of models and tasks, confirming its robustness. The authors also explained that the intervention strength is not sensitive to exact layer choices, ensuring flexibility across different architectures.

• Reviewer yxZC raised concerns about the real-world applicability of AGE and the stage-specific settings. The authors provided extensive domain-shift experiments showing AGE’s robustness across various out-of-domain benchmarks. They also clarified how the method’s stage-specific settings are chosen based on empirical evidence and how AGE performs well on both text-heavy and image-heavy domains.

**Reviewer Scores:**

I believe that if each reviewer had actively participated in the discussion, they would likely have made positive changes to their scores. Below is a brief analysis of why each reviewer might change their score:

• Reviewer s179: Given the thorough clarification on the layer-specific intervention and the non-universal positive gap in attention, the reviewer would likely recognize that the concerns were based on a misunderstanding. The clear explanation of the methodology could lead to a higher score.

• Reviewer Y1U5: After reviewing the detailed comparison with prior works, including the distinction between AGE and other attention-based methods (AGLA, DeCo, DeGF), and seeing the added explanations on hyperparameter selection and generalization, the reviewer would likely appreciate the novel contributions of AGE, leading to a score improvement.

• Reviewer sf9o: The reviewer initially raised concerns about sensitivity to layer choices and the method’s real-world applicability. However, the authors provided solid evidence of AGE’s robustness across models and tasks, including domain-shift benchmarks, which would likely convince the reviewer that the method is indeed flexible and generalizable, prompting a higher score.

• Reviewer yxZC: The reviewer expressed concerns about the practical applicability of AGE in real-world scenarios and the stage-specific settings. After seeing the domain-shift experiments and the added explanation on stage selection, the reviewer would likely recognize the flexibility of AGE, which would likely lead to a higher score.

---

### Decision · Program_Chairs · 2026-01-26

Accept (Poster)